



# Intercomparison of global ground-level ozone datasets for health-relevant metrics

Hantao Wang[1], Kazuyuki Miyazaki[2], Haitong Zhe Sun[3], Zhen Qu[4], Xiang Liu[5], Antje Inness[6], Martin Schultz[7], Sabine Schröder[7], Marc Serre[1], J. Jason West[1]

[1]Department of Environmental Sciences and Engineering, University of North Carolina at Chapel Hill, Chapel Hill, 27599, USA

[2]Jet Propulsion Laboratory, California Institute of Technology, Pasadena, 91125, USA

[3]Department of Paediatrics, National University of Singapore, Singapore, 117549, SG

[4]Department of Marine, Earth and Atmospheric Sciences, North Carolina State University, Raleigh, 27606, USA

[5]Department of Earth and Planetary Sciences, Harvard University, Cambridge, Massachusetts, 02138, USA

[6]ECMWF, Shinfield Park, Reading, RG2 9AX, UK

[7]Jülich Supercomputing Centre, Forschungszentrum Jülich, Jülich, 52428, Germany

*Correspondence to*: J. Jason West (jasonwest@unc.edu)

**Abstract**

Ground-level ozone is a significant air pollutant that detrimentally affects human health and agriculture. Global ground-level ozone concentrations have been estimated using chemical reanalyses, geostatistical methods, and machine learning, but these datasets have not been compared systematically. We compare six global ground-level ozone datasets (three chemical reanalyses, two machine learning, one geostatistics) against one another and relative to observations, for the ozone season

daily maximum 8-hour average mixing ratio, for 2006 to 2016. Results show significant differences among datasets in global average ozone, as large as 5-10 ppb, multi-year trends, and regional distributions. For example, in Europe, the three chemical reanalyses show an increasing trend while the other datasets show no increase. Among the six datasets, the population exposed to over 50 ppb varies from 60.8% to 99% in East Asia, 17% to 88% in North America, and 9% to 77% in Europe (2006–2016 average). These differences are large enough to impact health and other applications. Comparing with Tropospheric Ozone

Assessment Report (TOAR) II ground-level observations, most datasets overestimate ozone, particularly at lower observed concentrations. In 2016, across all stations, R² ranges among the six datasets from 0.35 to 0.63, and RMSE from 5.28 to 13.49 ppb. Performance further declines when considering only stations with observations above 50 ppb. Although some datasets share some of the same input data, we found important differences among these datasets, likely from variations in approaches, resolution, and other input data, highlighting the importance of continued research on global ozone distributions.

**1. Introduction**

Tropospheric ozone is a secondary pollutant that significantly impacts human health, plant life, and the climate system. Past studies have shown that ozone exposure can cause health effects ranging from mild subclinical symptoms to mortality (Balmes,



2022). The Global Burden of Disease 2021 (GBD) study estimated that ground-level ozone contributed to approximately 490,000 global deaths in 2021, representing 0.7% of all deaths that year (Brauer et al., 2024). Ozone exposure is harmful not

only to humans but also to plants. Ozone can enter plants through their stomata and cause oxidative damage, which reduce the global yields of major crops such as soybean, wheat, rice, and maize (Ainsworth, 2017; Mills et al., 2018a). Ozone is also an important greenhouse gas, ranking third behind carbon dioxide and methane in its contribution to anthropogenic climate change (Masson-Delmotte et al., 2021). Since the mid-1990s, tropospheric ozone has increased above 11 regions (Western North America, Eastern North America, Southeast North America, Northern South America, Northeast China/Korea, The Persian

Gulf, India, South east Asia, Malaysia/Indonesia, Europe, Gulf of Guinea) of the Northern Hemisphere, particularly for population-weighted ozone metrics (Gaudel et al., 2020). Using one global ozone dataset, from data fusion of ground observations and chemical model outputs, it is estimated that in 2017 21.3% of the global population was exposed to ozone concentrations above 65 ppb, and 96% lived in areas where concentrations exceeded the WHO guideline (30 ppb for annual metric) (Becker et al., 2023; Delang et al., 2021). However, the lack of knowledge of the ground-level ozone distribution limits

our ability to accurately assess ozone impacts on human health and crops.

The Tropospheric Ozone Assessment Report (TOAR) aggregates ozone observations from thousands of monitoring stations worldwide, forming the most extensive ground-level ozone monitoring data compilation to date (Schultz et al., 2017). Using the TOAR dataset, researchers have analyzed the global distribution, trends, and impacts of surface level ozone (Gaudel et al.,

2018). Currently, the second phase of the Tropospheric Ozone Assessment Report (TOAR-II) aims to include additional ground-based stations, especially new networks in China and India. However, despite significant progress, there remain large regions with limited ground-based monitoring, and a gap in understanding ground-level ozone variations over time and space. To bridge gaps in regions lacking ozone monitors, various methods, including chemical reanalysis based long-term data assimilation calculations, machine learning, and geostatistical methods have been employed. Chemical reanalysis is an

approach that integrates observations from various sources including satellites using data assimilation and chemical transport models (CTMs) to reconstruct historical atmospheric chemical compositions and understand long-term changes and trends in air quality and climate forcing (Miyazaki et al., 2020b). Tropospheric ozone records have been provided in recent chemical reanalyses including the Tropospheric Chemistry Reanalysis Version 2 (TCR-2, (Miyazaki et al., 2020b)), the Copernicus Atmosphere Monitoring Service (CAMS, (Inness et al., 2019)), and data assimilation using the GEOS-Chem joint model

(GEOS-Chem, (Qu et al., 2020b)). In addition, two machine learning estimates of global ground-level ozone have been produced to date: one using a space-time Bayesian neural network trained on TOAR observations and CMIP6 simulations (Sun et al., 2022), and another with a cluster-enhanced ensemble learning method that utilizes various data sources (Liu et al., 2022). Finally, geostatistical methods were applied by DeLang et al. who used Bayesian Maximum Entropy (BME) to estimate ozone through a data fusion of TOAR observations and output from multiple CTMs (Delang et al., 2021).  This approach was

further enhanced by incorporating the Regionalized Air Quality Model Performance (RAMP) framework to correct model biases (Becker et al., 2023). These estimates of global ozone distributions and trends have supported analyses of health impacts.



For example, ozone estimates of DeLang et al. were used in both the GBD 2021 study (Murray et al., 2020), and in a study of ozone health effects in urban areas globally (Malashock et al., 2022).

However, there remains a lack of knowledge regarding the consistency of ground-level ozone estimates, distributions, and long-term trends across these global ozone mapping products. Potential inconsistencies in these datasets could significantly impact public health research, especially in assessing the risks of ozone-related health, impacts and may impede the development of effective environmental policies and ozone management strategies (Post et al., 2012). Although each dataset incorporates a considerable amount of observational information and model simulations through various methodologies, each

inherently incorporates biases from these data sources during the fusion processes. While satellite measurements of precursor species can be used to constrain surface and lower tropospheric ozone in chemical reanalysis (Miyazaki et al., 2012), the performance of chemical reanalysis surface ozone is limited in part by the low sensitivities of satellite ozone measurements near the surface, as well as model simulation errors. Data fusion methods integrate outputs from multiple models with inherent biases, potentially propagating these biases to the final estimates (Delang et al., 2021). Furthermore, machine learning methods

trained on observation data may yield inaccuracies in rural and remote areas due to the uneven distribution of ground-level ozone monitoring stations (Liu et al., 2022; Betancourt et al., 2022). Therefore, conducting comparisons and evaluations of various types of ground-level ozone mapping products is essential to understand the inconsistencies and biases in these datasets, ultimately benefiting global health studies.

This study aims to compare ground-level ozone concentrations estimated by six datasets, and to evaluate their accuracy over the 2006-2016 period, with a particular emphasis on their capacity to represent long-term ozone trends across different regions. The comparison and evaluation include three chemical reanalysis datasets, two machine-learning datasets, and one geostatistical dataset. The period 2006-2016 is chosen as the period over which the six datasets all produce ozone estimates. The ozone seasonal daily maximum 8-hour average mixing ratio (OSDMA8) was selected as the health-relevant metric for

annual ozone evaluation (Turner et al., 2016). We employed a comprehensive set of metrics to assess the congruence between these datasets, globally and regionally, including for long-term population weighted ozone outdoor exposure. Relative to the latest TOAR-II observational dataset, this study also examines the six datasets' ability to estimate ground-level ozone concentrations across various regions for the years 2006-2016. This research endeavors to characterize differences among ground-level ozone datasets, including discrepancies in ozone estimates, distributions, and trends, that could hinder evaluation

of ozone's effects on health and agriculture, as well as impede the formulation of effective environmental policies. Although the primary focus of this study is on health impacts, the results are also largely applicable to agricultural and ecosystem impacts.



## 2. Data

As shown in Table 1, this study compares and evaluates ground-level ozone estimates from six global ozone mapping products in three categories. We utilized ozone seasonal daily maximum 8-hour average mixing ratio (OSDMA8) as the yearly ozone
metric across all datasets. OSDMA8 is defined here as the maximum of the six-month running monthly mean daily maximum 8-hr ozone (MDA8) from January of the current year wrapping to March of the following year (Delang et al., 2021). OSDMA8 is GBD's ozone metric for quantifying health effect from long-term ozone exposure (Brauer et al., 2024), and it is the metric used in the World Health Organization's air quality guidelines (Organization, 2021). All observations and model estimates are converted to OSDMA8 using the same algorithm. Details on the input data used to construct each dataset are available in the
Supporting Information (SI).

### 2.1 Geostatistical ozone dataset

The BME dataset uses geostatistical methods to provide high-resolution global ground-level ozone estimates. First, M³Fusion creates a composite of multiple global atmospheric chemistry models by weighting them based on their performance (Chang et al., 2019), then BME data fusion integrates this multi-model composite with observations in space and time, and finally
BME estimates are refined from $0.5° \times 0.5°$ to $0.1° \times 0.1°$ (Delang et al., 2021). The observations are from TOAR-I for 1990 to 2017, complemented by data from the Chinese National Environmental Monitoring Center (CNEMC) for 2013 to 2017. The latest version of this dataset employs RAMP for bias correction of M³fusion inputs (Becker et al., 2023). The BME ozone estimates are more accurate than the average outputs from multiple models, achieving an $R^2$ of 0.63 at $0.1° \times 0.1°$ resolution, as evaluated against observations through cross-validation (Delang et al., 2021). Furthermore, incorporating RAMP into the
BME process significantly improves $R^2$ by 0.15, especially in areas far from monitoring stations, as demonstrated through checkerboard cross-validation (Becker et al., 2023).

### 2.2 Machine learning ozone datasets

We utilized two machine learning global ground-level ozone datasets from the University of Cambridge, and Nanjing University. The University of Cambridge's machine learning (UKML) dataset was developed using a space-time Bayesian
neural network, fusing various data sources including historical observations, CMIP6 multi-model simulations (AerChemMIP historical simulations and ScenarioMIP projections), population distributions, land cover properties, and emission inventories (Sun et al., 2022). The UKML model categorized TOAR-I monthly ozone observations from 1990 to 2014 into urban and rural areas, and used these as labels for supervised learning. This model generates monthly global gridded ozone estimates from 1990 to 2019, downscaled to a $0.125° \times 0.125°$ spatial resolution. It exhibited great performance in predicting urban and rural
surface ozone concentrations, with $R^2$ values ranging from 0.89 to 0.97 and RMSE values between 1.97 and 3.42 ppb (Sun et al., 2022).



Nanjing University's machine learning (NJML) dataset was created using a cluster-enhanced ensemble machine learning method. This dataset integrates various data sources, including satellite observations, atmospheric reanalysis, land cover properties, emission inventories and meteorological features (Liu et al., 2022). The main input data for NJML include meteorological parameters from ERA5, atmospheric chemistry from the CAMS chemical reanalysis, aerosol concentrations from MERRA-2, satellite observations from OMI/Aura, and emissions data from CEDS, spanning from 2003-2019 with varying spatial resolutions. It utilizes the monthly mean of daily maximum 8 h average (MDA8) data from TOAR-I and CNEMC observations from 2003–2019 as training data. The NJML dataset produces monthly global gridded ozone estimates from 2003 to 2019 with a $0.5° \times 0.5°$ spatial resolution. The model demonstrates robust performance in both spatial and temporal predictions of ground-level ozone, with $R^2$ values of 0.909 and 0.925, respectively (Liu et al., 2022).

## 2.3 Chemical reanalysis products

We utilized surface ozone analysis fields from three chemical reanalysis products: the Tropospheric Chemistry Reanalysis Version 2 (TCR-2, (Miyazaki et al., 2020b)), the Copernicus Atmosphere Monitoring Service reanalysis (CAMS, (Inness et al., 2019)), and the GEOS-Chem reanalysis (GEOS, (Qu et al., 2020b)). Different from the machine learning and geostatistical ozone datasets, the chemical reanalysis products utilized satellite observations of atmospheric composition to produce three-dimensional profiles of atmospheric composition. In situ surface observations were not included in the global chemical reanalysis data assimilation. Because of the lack of direct observational constraints, challenges remain in estimating surface ozone in the current reanalysis products (Huijnen et al., 2020). Detailed comparisons of these reanalyses for ozone over the entire troposphere have been conducted by the TOAR-II chemical reanalysis working group (Sekiya et al., 2024; Jones et al., 2024), but without a focus on the ground level as analyzed here.

TCR-2 was generated by assimilating multiple satellite observations into the MIROC-Chem model, that was developed as a part of the multi-model multi-constituent data assimilation (Miyazaki et al., 2020a). The meteorological fields were nudged to the European Centre for Medium-Range Weather Forecasts (ECMWF) Interim Reanalysis meteorology. The data assimilation employed is an ensemble Kalman filter technique, which was used to effectively correct the emissions and concentrations of various chemical species (Miyazaki et al., 2020b). The assimilated data includes ozone, CO, $NO_2$, $HNO_3$ and $SO_2$ from satellite instruments such as OMI, MLS, GOME-2, SCIAMACHY and MOPITT over the period from 2005 to 2021. TCR-2 provides 2-hourly global ozone profiles at a $1.1° \times 1.1°$ spatial resolution, with the regional mean ozone bias against global ozonesonde measurements ranging from -0.4 to 4.2 ppb in the lower troposphere (850-500 hPa) (Miyazaki et al., 2020b).

CAMS, operated by the European Centre for Medium-Range Weather Forecasts (ECMWF) on behalf of the European Commission, provides the global reanalysis dataset on atmospheric composition developed by ECMWF. The main inputs for the CAMS ECMWF Atmospheric Composition Reanalysis 4 (EAC4) chemical reanalysis are retrievals of CO, Ozone, $NO_2$ and aerosol optical depth (AOD) from multiple satellite instruments including MLS, OMI, GOME-2, SCIAMACHY, MIPAS,



SBUV/2 and MOPITT, covering various periods ranging from 2003. CAMS employed the four-dimensional variational data assimilation (4D-Var) method to integrate the satellite measurements under ECMWF's Integrated Forecasting System (IFS) CB05 model (Inness et al., 2019). It provides 3-hourly global profiles of ozone and other species at a $0.75° \times 0.75°$ spatial resolution. While CAMS generally improves over previous analyses, challenges and biases remain, particularly at high
latitudes and in accurately capturing seasonal variations (Inness et al., 2019).

The GEOS-Chem dataset is developed through 4D-Var data assimilation of $NO_2$ column densities using the GEOS-Chem adjoint model that includes updates in stratospheric and halogen chemistry (Henze et al., 2007). The GEOS-Chem model is driven by the Modern-Era Retrospective analysis for Research and Applications, Version 2 (MERRA-2) meteorological fields
from the NASA Global Modeling and Assimilation Office (GMAO). Prior anthropogenic emissions of NOx, $SO_2$, $NH_3$, CO, NMVOCs (non-methane volatile organic compounds), and primary aerosols were obtained from the HTAP 2010 inventory version 2 (Janssens-Maenhout et al., 2015). Operating at a $2° \times 2.5°$ resolution, the assimilation estimates global ozone more accurately than the forward model from 2006 to 2016 by deriving emissions of $NO_2$ through inverse modelling. The GEOS-Chem dataset exhibits a small bias across all ozone metrics, and among metrics it has the best spatial consistency for MDA8
($R^2 = 0.88$) (Qu et al., 2020b). However, the model has limitations in accurately capturing regional variations and seasonal trends in ozone concentrations.

## 2.4 Ground-level ozone observations

For the evaluation in this project, we utilized both urban and non-urban ground-level ozone observations for the yearly OSDMA8 metric from the updated TOAR-II dataset, covering 2006 to 2016. This dataset represents the most extensive
collection of tropospheric ozone measurements available globally. Compared to TOAR-I (Schultz et al., 2017), TOAR-II incorporates an expanded dataset of ozone observations, notably including monitoring data from approximately 1,400 stations across China for the years 2015 to 2016 that are included in TOAR-II (https://toar-data.fz-juelich.de/gui/v2/dashboard/, last access: 15 November 2024). We require that at least 75% of the days in a month must have valid DMA8 values for that month to be included in the annual data calculations. The total number of observation sites used in our assessment varied from a
minimum of 3715 in 2006 to a maximum of 7013 in 2016. Given that three ozone products in this study utilize the TOAR-I dataset for training or input, evaluations using the latest TOAR-II dataset for sites not included in TOAR-I can provide more objective results. Figure S1 illustrates the spatial distribution of TOAR-II monitoring stations in 2016. The version of the TOAR-II database employed in this analysis, as of November 2024, may not represent its final version.

## 2.5 Population data

We analyzed ozone population exposures for each dataset using the globally gridded population data for the year 2019 from the Global Burden of Disease (GBD) 2019, which has a resolution of $0.1° \times 0.1°$ (Lloyd et al., 2019). Since we use the same gridded population data for all years of the project, we focus on differences in exposure attributable to changes in ozone levels





rather than changes in population. Therefore, population-weighted ozone over 2006 to 2016 can be biased even if the ozone data are unbiased.

## 3. Methodology

### 3.1 Long-term exposure comparison

Before comparing concentration estimates between datasets, we converted all ozone estimates from each dataset to OSDMA8, ensuring only one ozone estimate value per year for each grid cell (see the original temporal resolution in Table 1). Subsequently, we re-gridded all datasets to $0.1° \times 0.1°$ resolution to facilitate comparison at the same spatial scale. During re-gridding, we ensure that the average value of the finer grid cells matches that of the original coarse grid cell; for example, if a grid cell has a value of 30 ppb, then after re-gridding to finer grid cells, the average value of these grid cells will still be 30 ppb. Data over the ocean were excluded, retaining only land and populated islands for analysis. We calculated the yearly ozone for each dataset weighted by population and by area. We also regressed population-weighted mean ozone concentrations in different world regions of each dataset against the year to evaluate ozone long-term variations. For each grid cell we calculate the mean and standard deviation of the six OSDMA8 values obtained from each dataset to highlight regional differences and similarities. We also calculated the deviation from the ensemble mean for each dataset to assess geographic distribution variations. Furthermore, we compared ozone exposure differences in various regions for each dataset to evaluate the potential for health impacts. Here we estimate exposure as the ambient concentration in $0.1° \times 0.1°$ grid cells related to population at their residences, not including other factors that affect human exposure such as time-activity patterns. Details of these calculations are available with parameters in the SI. We use regional groupings defined by HTAP2 (Koffi et al., 2016), as detailed in the Table S7.

### 3.2 Pairwise spatial similarity comparison

We employed two quantitative metrics to classify how the datasets relate with one another: the Pearson correlation coefficient (R) and the root mean square difference (RMSD). The pairwise correlation R indicates the similarity in geographical distribution of ozone concentrations, and the RMSD quantifies the difference in ozone estimates between datasets. A higher R value suggests greater similarity in the spatial pattern between two datasets and a smaller RMSD indicates a less significant discrepancy in ozone concentration estimates between two datasets. We then group the datasets, adopting a method that maximizes the difference between the correlation R within and outside the groups; more details of the calculation can be found in Text S1.





## 3.3 Evaluation with ground-level observation

We utilized OSDMA8 from TOAR-II observations covering 2006 to 2016 to evaluate the six datasets. During the evaluation process, we retained the original resolution of the six datasets (Table 1). We adopted a point-to-grid evaluation approach, where the data from each TOAR-II observation site was matched with a corresponding grid cell in each dataset. For grid cells with a TOAR-II observation but no valid estimate in a dataset (NA value), we used the nearest valid estimate instead. This method ensures the same sample sizes for evaluation across all datasets while accommodating their varied resolutions and avoiding the challenge of gridding TOAR-II observations. Our point-to-grid evaluation approach ensures a consistent sample size and captures penalties for missing data in datasets. We assessed the performance of each dataset using the coefficient of determination ($R^2$) between ozone estimates and observations, and root mean square error (RMSE) as the primary metrics.

## 4. Comparison between ozone mapping products

### 4.1 Temporal trends

Both the area-weighted and population-weighted mean trends of global OSDMA8 reveal substantial differences among global ozone mapping datasets (Fig. 1). Notably, BME and CAMS have lower ozone values than other datasets, for both metrics, while UKML and NJML have higher ozone estimates, with differences between these datasets exceeding 5 ppb. The higher values in GEOS-Chem and TCR-2 may be attributed to the remaining high bias in the forecast models, which is commonly found in CTMs (Travis and Jacob, 2019). The population-weighted mean is higher than the area-weighted mean, by 5-10 ppb across all datasets, indicating higher ozone in more populated regions, and for UKML and BME, the disparity between population-weighted and area-weighted ozone concentrations appears to widen over time. Most datasets show an increase in global population-weighted mean ozone, but for the area-weighted ozone, the upward trend is reduced or there is no clear trend, depending on the dataset. This difference seems to suggest that ozone is increasing most in regions with high population. For the population-weighted mean trend, the Mann-Kendall trend test indicates an increase in ozone levels across five datasets over their respective coverage periods. The exception is NJML, which differs from the other datasets by showing a clear decreasing trend for both the area-weighted and population-weighted ozone, shown in the Mann-Kendall trend test with very high certainty. Increasing ozone trends are consistent with ground-based ozone measurements in the Northern Hemisphere from 2005 to 2016 (Gaudel et al., 2018; Fleming et al., 2018).

Fig. 2 illustrates regional ozone changes per decade, weighted by population, across different regions in each dataset over their respective time frames. CAMS, with lower concentrations (Fig. 1), shows the largest regional ozone rise, with most global regions experiencing increases, particularly in the Sub Saharan/Sub Sahel Africa and Middle East regions, exceeding 8 ppb per decade. NJML, despite its overall decreasing trend in Fig. 1, does not uniformly show declines across all regions (Fig. 2). The decrease in NJML is predominantly in North America, notably over 8 ppb per decade in the US and Canada, and Europe,



while East Asia, Sub Saharan Africa, and South America exhibit increases. BME and UKML, with the longest duration, both display decreasing trends in North America, and Europe, and increases in South Asia, East Asia, and Middle East. Both datasets indicate greater decreases in North America than in Europe and more significant increases in South Asia than in East Asia and Middle East. However, BME shows a downward trend in Mexico and Central America and an upward trend in Northern Africa, while UKML exhibits the reverse. TCR-2, NJML, and CAMS, having similar years of coverage, show less pronounced trends in some regions compared to the other datasets. TCR-2's trends in Fig. 2 are less distinct, except for the decrease in North America and the increase in South Asia, mirroring those of GEOS-Chem, which exhibits the least decadal ozone change, likely due to its shortest coverage span and not directly assimilating ozone from satellite observations. In East Asia, all six datasets indicate an upward ozone change, while South Asia shows an increase in all but NJML. In North America, all datasets except CAMS display a downward trend, and in Europe, BME, NJML, and UKML show a decline, contrasting with increases in the three chemical reanalysis datasets. Previous analyses using TOAR observations indicate that from 2000 to 2014, most of North America and some European sites experienced significant negative trends, while many sites in East Asia exhibited significant positive trends (Fleming et al., 2018; Gaudel et al., 2018; Chang et al., 2017). These observed trends in North America, Europe and East Asia seem to agree best with the trends estimated by BME and UKML.

## 4.2 Difference maps

Fig. 3 shows the spatial maps of the 11-year (2006-2016) average of the annual multi-model means of OSDMA8 from the six datasets, and the associated standard deviations. India, China, and the Middle East are estimated to have the world's highest average ozone concentrations in the multi-model average. over 50 ppb. High ozone levels are also found in parts of Europe and the eastern United States. Notably, regions in southern Africa near the Atlantic Ocean emerge as primary areas of ozone pollution, where some locations have average concentrations exceeding 60 ppb. Conversely, the Amazon Basin in South America, Central Africa, and Canada exhibit relatively lower ozone concentrations, below WHO 30 ppb guideline. The six datasets show greater variation (high standard deviations above 10 ppb) in South America and Africa, particularly in rainforest regions, compared to North America and Europe, notably since these regions lack ozone monitors. The eastern coast of China also exhibits significant discrepancies with standard deviations above 15 ppb. Fig. 4 compares the mean ozone concentration for each dataset with the multi-dataset average (Fig. 3(a)), showing wide variation in the magnitude and spatial distributions of ozone concentrations among the datasets. BME and CAMS display lower values than the average of six datasets in most regions, consistent with Fig. 1. BME records concentrations higher than average in central South America and central Africa near the Atlantic, while CAMS shows elevated levels in Southeast Asia and along the Middle East coast, contrasting TCR-2's lower coastal and higher inland concentrations. NJML and UKML report above-average values, except for NJML in southern China and UKML near the Sahara Desert and the Indian Ocean.





## 4.3 Pairwise spatial similarity

We calculated the correlation and RMSD between each pair of datasets for each year from 2006 to 2016. Fig. 5 displays the average correlation and RMSD values over these 11 years as heat maps. Fig. 5(c) presents a scatter plot of the correlations and RMSD for each dataset pair. Using the correlation heatmap (Fig. 5(a)), we categorized the six datasets by the maximum
difference method, identifying NJML as a distinct group (Group B) and the other five datasets as Group A. NJML's separation indicates its significant divergence in ozone geographic distribution compared to others. The scatter distribution in Fig. 5(c) reveals that most Group A data points cluster in regions of high correlation and low RMSD, suggesting broadly consistent ozone geographic distribution and concentration estimation within this group. Nevertheless, there are still substantial disagreement among the current reanalysis products, likely because of the differences in forecast model performance and data
assimilation configuration. Conversely, Group B has lower correlations. Interestingly, RMSD does not consistently decrease with increasing correlation, indicating that similar geographic distribution patterns can still yield significant differences in ozone concentration estimates. This is particularly evident with CAMS and GEOS-Chem, which exhibit the highest correlation with a large RMSD, suggesting substantial differences in ozone estimation.

## 4.4 Long-term ozone exposure

Fig. 6 illustrates the distribution of population in various regions exposed to average OSDMA8 from 2006 to 2016, as per each dataset. Detailed plots of population exposure for each year (2006 to 2016) are shown in Figure S10. For the period 2006-2016, a majority of the population in most datasets is exposed to concentrations above 50 ppb. Populations in regions such as East Asia and South Asia appear to be exposed to higher ozone concentrations in all datasets compared to other regions. Conversely, populations in the Sub Saharan Africa and Southeast Asia regions typically experienced concentrations below 50
300    ppb. The different regions show different distributions of population ozone exposure, and comparisons between datasets reveal considerable variations in the ozone distribution for each region. Some datasets (e.g., CAMS and TCR-2) show a wider distribution of population across ozone concentrations compared to others (e.g., NJML). In BME and CAMS, after South Asia, a significant fraction of the population in the East Asia region is exposed to levels above 50 ppb, while this proportion in North America, Europe, and the Middle East is less than in the other four datasets. When focusing on exposure above 70 ppb, South
Asia dominates in BME, CAMS, and NJML, while East Asia leads in GEOS-Chem, UKML, and TCR-2. All six datasets clearly demonstrate a higher impact of ozone pollution in Asia compared to North America and Europe, aligning with previous findings based on TOAR observations (Chang et al., 2017). Table 2 elucidates each region's population share above different ozone concentration levels. For BME and CAMS, the global average of the population exposed to more than 50 ppb is 42.5% and 48.1%, respectively, indicating that more than half of the population us exposed to lower concentrations. Regional
exposure estimates vary in East Asia, where the proportion of the population exposed to more than 50 ppb ranges from 60.8% in BME to over 90% in UKML, GEOS-Chem, and TCR-2. The differences are stark in Europe, with BME and CAMS showing only 16.1% and 9.5% exposure, respectively, over 50 ppb, while NJML, UKML, and TCR-2 report over 70%. TCR-2 and





UKML project notably higher exposures in East Asia, with 40.9% and 31.4% of the population exposed to levels above 70 ppb, respectively. In the Middle East, TCR-2's estimates are significantly higher than other datasets, indicating that 38.1% of the population is exposed to average concentrations above 70 ppb.  The six datasets agree that a large majority of the global population is exposed to ozone above the WHO guideline for OSDMA8 (30 ppb) with percents ranging from 93% (CAMS) to 99% (NJML).

## 5. Evaluation against TOAR-II observations

### 5.1 Evaluation of ground-level ozone in 2016

We conducted regression and bias analyses for each dataset in comparison with TOAR-II observations for each year from 2006 to 2016. Fig. 7(a) illustrates the scatterplot from the linear regression analysis of each dataset against the 7013 TOAR-II observations in 2016, accompanied by a density core that visualizes the data point distribution. The year 2016 is presented here because it has the highest number of TOAR-II observations from 2006 to 2016, and other years can be found in Figure S7. For 2016, BME outperforms other datasets, with the highest $R^2$ (0.63) and lowest RMSE (5.28 ppb), its density core intersecting the y=x line. BME has an advantage in that its methods should nearly match the observed values for locations used as inputs to the data fusion.  Consequently, we conduct another validation for TOAR-II sites not used as input for BME in 2016 (Figure S13). After excluding all sites previously used as BME input, using 3134 observations for validation, BME performs well after the exclusion of input sites, evidenced by a minor decrease in $R^2$ to 0.51. In Fig. 7(a), all three chemical reanalysis datasets exhibit a relatively good $R^2$ ranging from 0.35 to 0.41, comparable to the performance of the machine learning datasets, which have $R^2$ values of 0.37 and 0.38. Among these five datasets, CAMS has the lowest RMSE (7.59 ppb), which is better than other chemistry reanalysis products but relatively low $R^2$ (0.35). Its density core slightly above the y=x line suggests CAMS estimates are marginally lower than TOAR-II observations. GEOS-Chem and TCR-2 demonstrate adequate performance, albeit with higher RMSE values of 10.27 ppb and 13.23 ppb, respectively. Their density cores positioned below the y=x line indicate that these models tend to produce higher estimates compared to the TOAR-II observations. NJML, despite differing geographic distributions from other datasets (Fig. 5), shows acceptable performance with higher $R^2$ (0.38) than CAMS and lower RMSE (8.63) than TCR-2. UKML exhibits the highest RMSE of 13.49 ppb, and its density core region rarely intersects the y=x dashed line, indicating a significant overestimation.

Fig. 7(b) focuses only on TOAR-II sites with OSDAM8 value above 50 ppb, showing that $R^2$ is reduced compared to the comparison of all ozone measurements (Fig. 7(a)) for all six datasets, suggesting overall weaker prediction accuracy at higher concentrations. All six datasets show decreasing performance from BME, NJML, and UKML to TCR-2, GEOS-Chem, and CAMS, with $R^2$ of 0.37, 0.3, 0.26, 0.25, 0.17, and 0.07, respectively. From density cores in Fig. 7(b), UKML, GEOS-Chem and TCR-2 are generally below the y=x dashed line, suggesting that overestimation of higher ozone concentrations is more prevalent than underestimation across these datasets. In contrast, the density cores of BME and CAMS in Fig. 7(b) lie above



the dashed line, implying their estimates are lower than observed values at high concentrations, with BME showing a larger

deviation compared to its position in Fig. 7(a). However, NJML's density core is closer to the y=x dashed line under high-

concentration conditions, indicating closer alignment with observed values. Fig. 8 shows the normalized mean bias for

stratified concentration intervals in 2016, which provides insights into the average discrepancy between estimates and TOAR-

II observations across ozone concentration ranges. All six datasets overestimate TOAR-II observations below the 40%

concentration interval. Only BME underestimates above the 40% concentration level, CAMS underestimates above the 80%

concentration interval, and NJML underestimates above 90% concentration interval, aligning with the observations presented

in Fig. 4. BME demonstrates the smallest mean bias, particularly below the 50% concentration level and CAMS shows the

smallest mean bias from 50% to 90% concentration interval. From 90% to 100% concentration interval, NJML and GEOS-

Chem have the smallest mean bias. In summary, BME and CAMS perform better overall in terms of normalized mean bias,

with other models tending to overestimate ozone at almost all concentrations.

**5.2 Evaluation of ground-level ozone in different countries**

Table 3 presents the validation results for different countries or regions using TOAR-II observations in 2016, focusing on the

countries with the highest number of sites. Here we use the $R^2$ to assess the strength of the spatial correlation and RMSE to

measure the bias across each country or region. The performance of each dataset varies by region, indicating that a dataset's

overall performance does not guarantee its effectiveness in all regions. Reasonable $R^2$ and RMSE values are seen across all 6

datasets in the United States; BME leads with the highest $R^2$ (0.71) and lowest RMSE (4.12 ppb), and TCR-2 has the lowest

$R^2$ (0.23) with highest RMSE (10.58 ppb). In Japan, BME leads with an RMSE of 4.59 ppb, followed by CAMS at 4.95 ppb,

and UKML has the highest RMSE (18.25 ppb). Although there are over 1000 monitors in Japan, all datasets show poor $R^2$

values below 0.1. The six datasets also perform poorly in South Korea, where TCR-2 has the highest RMSE (18.53 ppb), BME

has the lowest RMSE (7.33 ppb). The performance of datasets within China exhibits significant variability, where BME and

NJML demonstrate relatively good performance, and CAMS exhibit poor performance for $R^2$, while for RMSE, CAMS

performs better than GEOS-Chem, TCR-2 and UKML. For other countries, which serve as a test of model performance in

areas with sparse observations, all datasets exhibit better $R^2$ and RMSE values than in South Korea, with TCR-2, NJML and

BME demonstrating particularly better performance than others. Overall, BME demonstrates strong performance in most

countries, particularly in the United States, where it achieves the highest $R^2$ and the lowest RMSE, suggesting both strong

spatial correlation with TOAR-II observations and high accuracy. NJML exhibits mixed performance, with relatively high $R^2$

values indicating good correlation in the United States and China, but it falls short in EU-27 and Canada with high RMSE and

low $R^2$. UKML presents consistently high RMSE values across countries suggesting high bias. CAMS displays variable

performance with low $R^2$ values in China, indicating a lack of spatial correlation, yet its RMSE values are relatively small

across all regions when compared to other chemical reanalysis datasets. GEOS-Chem and TCR-2 exhibit reasonable spatial

correlations in Europe, the United States, China, and Canada. Notably, they outperform all other datasets in Canada, except

for BME. TCR-2 demonstrates the best $R^2$ performance in other countries with less monitoring data. However, TCR-2 also




presents high RMSE values across all regions. All six datasets exhibit lower spatial correlation compared to TOAR-II observations in countries with high monitoring density, such as Japan and South Korea, than in countries with lower monitoring densities. NJML, UKML, GEOS-Chem and TCR-2 show overestimates compared to the TOAR observations in every country in the table. Extending the analysis to the period from 2006 to 2016 (see tables in SI), the percentage of underestimates from 6 datasets compared to TOAR observations in all countries is below 20%.

## 5.3 Evaluation of ground-level ozone across different years

Fig. 9 presents time series plots of $R^2$ and RMSE from the evaluation of each database against TOAR-II observations from 2006 to 2016. It is important to note that the years 2015 and 2016 include observations from China. BME consistently shows the largest R², indicating its robust performance near the monitor locations due to the utilization of observational data as input and the effective exploitation of spatiotemporal autocorrelation among stations. Apart from BME, NJML outperforms other datasets in R² from 2010 to 2015, TCR-2 leads in 2007 and 2016, while UKML does so in 2008 and 2009. Five datasets, excluding NJML, demonstrate a drop in $R^2$ in 2010. All datasets show an increase in $R^2$ from 2015 to 2016. BME maintains the lowest RMSE throughout the period, indicating the most accurate predictions. CAMS also performs good in terms of RMSE. GEOS-Chem consistently has lower RMSE than both TCR-2 and UKML. Meanwhile, NJML exhibits a decreasing RMSE trend from 2006 to 2016. The clear differences in time series of RMSE correspond with the yearly mean trends in Fig. 1. Datasets with lower ozone values, BME and CAMS, also exhibit lower RMSE, whereas those with higher estimates, specifically TCR-2 and UKML, have higher RMSE.

## 6. Discussion

From the comparison, we find there are large differences in ozone concentration estimates among datasets. Figure 1(b) illustrates that BME and CAMS report lower ozone estimates compared to UKML and NJML, with differences exceeding 5 ppb. NJML demonstrates a decreasing trend in population-weighted yearly mean, while the five others show an increasing trend. Divergence among datasets becomes even more evident in the analysis of regional ozone trends (Fig. 2). CAMS records that ground-level ozone is rising more than 8 ppb per decade in sub-Saharan Africa and the Middle East, while NJML indicates a general downward trend in these regions. Moreover, the ozone concentrations decreased in Europe according to BME, NJML, and UKML, yet increase across all three chemical reanalysis datasets. Differences in regional distributions lead to variability in exposure estimates. Among the six datasets, the population exposed to more than 50 ppb of ozone in Europe from 2006 to 2016 spans a broad range, from as low as 9% for CAMS to over 70% for NJML, UKML, and TCR-2. This highlights the importance of removing systematic biases from these data sets before applying them to exposure estimates. In East Asia, exposure levels are consistently higher, with the percentage of the population affected ranging from 60.8% for BME to more than 90% for UKML, GEOS-Chem, and TCR-2 based on average OSDMA8 data over the same period. Global average




exposures also vary, with the proportion of the population exposed to more than 50 ppb ranging from 42% to 70% across the six datasets.

410

Despite notable disparities in estimates, we still find some regional and temporal similarities across the six datasets. An overall upward trend in ozone concentrations is evident across most datasets, particularly when examined as population-weighted means. In Fig. 3(a) high ozone concentrations are predominantly found in regions with elevated anthropogenic and industrial emissions, while forests and sparsely populated areas have lower ozone concentrations, consistent with findings based on observations (Mills et al., 2018b; Fleming et al., 2018). In Fig. 3(b) the standard deviation among six datasets is high in part of South America and Africa, especially in the rainforest areas, probably because of the lack of observational data in these areas and uncertainties in the emissions inventories (Pfister et al., 2019). However, for most regions it is low, such as North America and South Asia, indicating a good level of agreement on ozone estimates. The high pairwise correlation in Fig. 5(a) supports that the geographical distributions of ground-level ozone are similar among most of datasets. The histograms of ground-level ozone exposure among the population (Fig. 6) reveal the shared characteristic of widespread high ozone exposure in East and Southeast Asia (Fleming et al., 2018).

When evaluating datasets against the TOAR-II observations, differences in performance are seen among six datasets. BME performed well in the TOAR-II evaluation (Fig. 9), with minimal mean bias below the 50% concentration threshold (Fig. 8). Unlike the other databases, BME tends not to overestimate over the range of concentration, with a small underestimation bias. Still, after removing 3879 TOAR sites that were used as inputs to BME (Fig. S13), BME's performance remains robust, as seen by small shifts in RMSE (from 5.28 to 5.14) and $R^2$ (from 0.63 to 0.51). NJML and UKML, both utilizing TOAR-I as a training set, showed overestimation in most areas (Table 3). Despite NJML's distinct spatial distribution in Fig. 5, its validation results are comparable to other datasets. NJML exhibits a higher $R^2$ from 2010 onward, especially at high ground-level ozone concentrations (above 50 ppb), where prediction accuracy generally declines across all datasets. However, NJML has missing data in some coastal regions, particularly in European coastal countries, which may contribute to its elevated RMSE in Europe compared to other datasets in Table 3, since missing data are substituted with the nearest model grid cell. UKML's performance after 2010 is not as good as NJML and is worse than the chemical reanalysis datasets in 2011. CAMS, GEOS-Chem and TCR-2 primarily rely on satellite data, suggesting that they might not compare favorably with other datasets that used observations as input or training data. Despite this, CAMS unexpectedly outperforms the machine learning datasets in RMSE over the full year, especially for high ozone concentrations (50% to 90% range). In addition, as shown in Fig. 8, TCR-2, GEOS-Chem, NJML, and UKML all have widespread overestimation.

There are several possible explanations for the differences among the datasets, including several factors related to the characteristics, methodologies and input data for each dataset. BME has an unfair advantage in that it nearly matches observations at monitoring location. But as mentioned earlier, BME still shows superior performance after removing its



training data from the evaluation. BME's use of temporal autocorrelation to predict ozone in years where measurements are missing may help its good performance (Delang et al., 2021). The differing yearly ozone population-weighted mean trend in NJML compared to other datasets may be due to its unique input data, including land cover and satellite observations (Liu et al., 2022). The missing data near coastlines in NJML and relatively coarse resolution likely contribute to poorer performance in EU-27. For three chemical reanalysis datasets, previous studies have shown that significant challenges remain, particularly with respect to the representation of ozone in the lower troposphere, because of the limited sensitivity of satellite observations to ozone in the lower layers (Huijnen et al., 2020). Because of the lack of direct observational constraints at the surface in the chemical reanalyses, the better performance of CAMS may be attributable to the finer resolution, that enable better representation of small-scale ozone distribution features than the other reanalysis datasets, and also to the better performance of the forecast model to predict surface ozone. Nevertheless, the assimilation of precursor measurements provides important constraints, particularly with respect to the spatial gradient and temporal variation of ground-level ozone. The low RMSE of GEOS-Chem compared to UKML and TCR-2 might be because it shares the same data assimilation method with CAMS (Qu et al., 2020a). Moreover, TCR-2, GEOS-Chem, and CAMS perform well in the United States, Canada and EU27, which may be because these regions have well-established emissions inventories for modeling (Schmedding et al., 2020) and because data assimilation is used to estimate key precursor emissions from satellite observations in TCR-2 and GEOS-Chem. Optimizing additional precursor emissions, such as VOCs, from satellite observations is considered to be important to better represent surface ozone (Miyazaki et al., 2019; Sekiya et al., 2024; Miyazaki et al., 2012). The poor performance in South Korea and Japan could be because the coarse resolution models may not accurately capture ozone gradients in a nation with a high density of monitors (Punger and West, 2013; Sekiya et al., 2021). This suggests a need for continued efforts to improve the mapping resolution to capture spatial variability in these regions. Since most of the current reanalysis products still suffer from large systematic errors in their surface ozone analysis, it might be important to apply bias corrections while maintaining the detailed spatial and temporal variability of the original data using methods such as machine learning (Miyazaki et al., 2024) before performing exposure estimates. While these factors may help to explain differences between the datasets, we have not systematically tested them, and as discussed by Sekiya et al. (2024) and Jones et al. (2024), further inter-comparisons of reanalysis products and detailed discussions for improvement are required.

Although we conducted a comprehensive comparison and evaluation, this study still has some limitations. First, the comparison only focuses on land and inhabited islands, because of the focus on ground-level ozone impacts on health. Our estimates of population exposure are based on ambient concentration in each grid cell, ignoring other factors that impact ozone exposure, such as indoor ozone concentration. Also, using OSDMA8 as the metric to evaluate datasets might hide differences in model performance at hourly temporal resolution, which would need to be analyzed in a separate study. In instances of missing model estimates, we default to the nearest valid estimate to evaluate with TOAR-II observations. For datasets with coarse spatial resolution, this method may increase bias by double counting.





## 7. Conclusions


This study evaluates the consistency and accuracy of six ground-level ozone mapping products, developed using different methods. Substantial discrepancies among datasets are reflected in global and regional ozone trends, the spatial distribution of ozone, population exposure estimates, and model performance. The global population-weighted average has a maximum span of 10 ppb among the six datasets. In terms of long-term trends, BME, UKML, CAMS, and TCR-2 show a consistent upward

trend globally, while NJML shows a downward trend. Regionally, five datasets except CAMS show a downward trend in North America, and all six datasets demonstrate an upward trend in East Asia; In Europe, BME, UKML, and NJML report a downward trend, while the three chemical reanalysis datasets reveal an upward trend that is not seen in observations. These differences among datasets are sufficiently large that assessments of health impacts of ozone would differ significantly when using different ozone datasets. Model performance evaluation based on TOAR-II observations varied; in 2016, $R^2$ values

ranged from 0.35 to 0.63, and RMSE values ranged from 5.28 ppb to 13.49 ppb for all stations. BME performs well near monitoring locations with good $R^2$ and small RMSE. All five datasets, except for BME, exhibit similar $R^2$ values in 2016. NJML performs well after 2010 and shows robust performance under high ozone concentrations. Before 2010, UKML performs well, but after 2010, UKML shows decreased performance. Machine learning datasets tend to overestimate. The chemical reanalysis datasets perform comparably with the geostatistical and machine learning datasets, which is somewhat

surprising given that they were not designed to estimate ground-level ozone accurately and do not use ground-level observations as input. CAMS performs the best among the chemical reanalysis datasets in term of RMSE, although CAMS has difficulty capturing TOAR-II observations in China. In regions where TOAR-II observations are sparse, all datasets show RMSE values about 10 ppb, highlighting the difficulty in mapping ground-level ozone distributions in regions with little observational data. Conversely, in some regions with very dense TOAR-II observations, all datasets show $R^2$ values below 0.1,

highlighting the necessity for fine resolution mapping to accurately capture spatial variability.

Given that some of the datasets used similar input data, it is somewhat surprising to find the large discrepancies shown here, suggesting that applications of these datasets to health burden assessments, epidemiology or similar applications for agricultural and ecosystem impacts may differ strongly based on the dataset selected. More research will be needed before

different methods converge on similar estimates. Such research can include more widespread ground observations, improved used of satellite observations, improved chemistry-climate modelling, and further development of different data fusion methods. Also, it is not clear whether differences among different datasets are due mainly to the methods used or to differences in input data. In addition, establishing a formal benchmark test based on the evaluation methods described in this study for the yearly OSDMA8 metric is essential. This would allow for new mapping products to be easily assessed. The general findings

here of poor agreement among datasets may also be applicable to other air quality datasets or even datasets from other Earth system domains. According to this study, there is no clear consensus on the best ozone mapping methods. To further improve these ozone mapping products, researchers must update and adjust their methods and input data regularly and iteratively.



## 8. Code and data availability

Observational data are publicly available from the TOAR-II data portal (last accessed on 15 November 2024, toar-data.org).
The BME dataset of global ground-level ozone estimates (Becker et al., 2023) is publicly available at zenodo.org/records/10498857. The NJML dataset is publicly available at doi.org/10.5281/zenodo.6378092. The CAMS reanalyses data (Inness et al., 2019) are publicly available from https://ads.atmosphere.copernicus.eu/datasets/cams-global-reanalysis-eac4. The TCR-2 reanalyses data (Miyazaki, 2024) are publicly available from https://disc.gsfc.nasa.gov/datasets/TRPSCRO32H2D_1. Other datasets of global ozone concentrations can be obtained by
contacting the creators of these datasets.

## 10. Author Contributions

This research was conceived by HW, JJW, and MLS. HW, KM, HZS, ZQ, XL, and AI provided ozone concentration datasets. MS and SS provided TOAR-II observational data. Data analyses and numerical results were generated by HW with input from MLS and JJW. HW, MLS and JJW wrote the paper and all authors provided edits and comments on drafts.

## 11. Competing Interests

Some authors are members of the editorial board for *Atmospheric Chemistry & Physics*. The authors declare that they have no other conflicts of interest.

## 12. Financial Support

We gratefully acknowledge support for this work through National Aeronautics and Space Administration (NASA) grants
#NNX16AQ30G and #80NSSC23K0930. Part of this work was conducted at the Jet Propulsion Laboratory, California Institute of Technology, under contract with NASA. We also acknowledge the support of the NASA Atmospheric Composition: Aura Science Team Program (19-AURAST19-0044), Atmospheric Composition Modeling and Analysis Program (22-ACMAP22-0013), NASA Earth Science U.S. Participating Investigator program (22-EUSPI22-0005).






**Table 1: Overview of six global ozone mapping products.**

| Global ozone dataset | Model type | Resolution | Period | Temporal Resolution |
|---|---|---|---|---|
| Bayesian Maximum Entropy Data Fusion (BME)(Delang et al., 2021) | Geostatistics | $0.1° \times 0.1°$ | 1990-2017 | OSDMA8 |
| Cluster-Enhanced Ensemble Learning (NJML)(Liu et al., 2022) | Machine Learning | $0.5° \times 0.5°$ | 2003-2019 | Monthly MDA8 |
| Space-Time Bayesian Neural Network Downscaler (UKML)(Sun et al., 2022) | Machine Learning | $0.125° \times 0.125°$ | 1990-2019 | Monthly MDA8 |
| Copernicus Atmosphere Monitoring Service (CAMS)(Inness et al., 2019) | Chemical Reanalysis | $0.75° \times 0.75°$ | 2003-2020 | 3-Hourly |
| GEOS-Chem (GEOS)(Qu et al., 2020b) | Chemical Reanalysis | $2° \times 2.5°$ | 2005-2016 | MDA8 |
| Tropospheric Chemistry Reanalysis Version 2 (TCR-2)(Miyazaki et al., 2020b) | Chemical Reanalysis | $1.125° \times 1.125°$ | 2005-2020 | 2-Hourly |






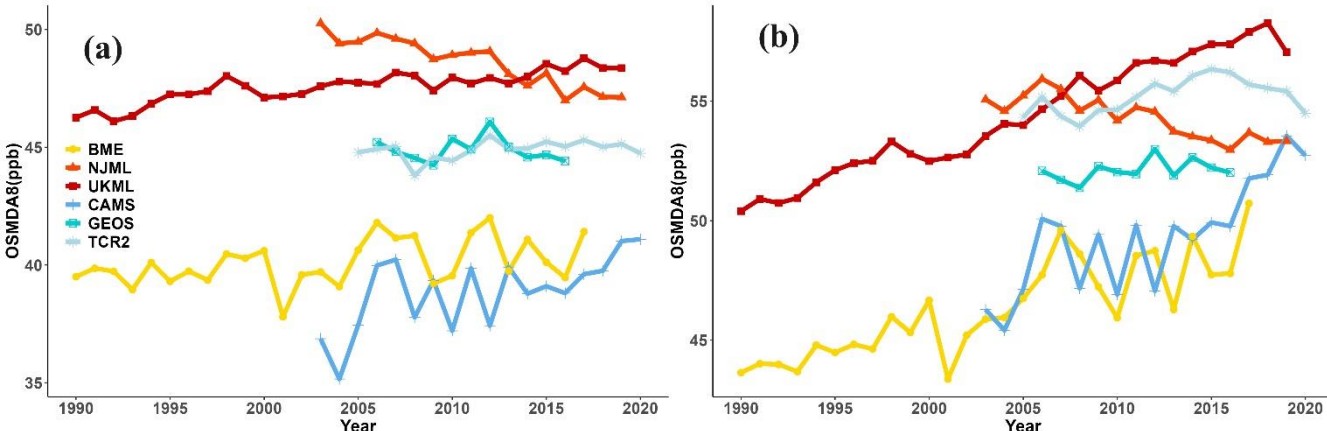

**Figure 1: Yearly trends of ground-level ozone for six datasets, shown for (a) the area weighted global mean ozone over land, and (b) population weighted global mean ozone, where ozone is expressed as OSMDA8. Yearly trends for individual world regions are shown in Figures S2 and S3. Mann-Kendall trend test for population weighted global mean over the full time series for each dataset: BME**

**0.688 ppb yr$^{-1}$ trend with p-value < 0.0001, NJML -0.691 ppb yr$^{-1}$ with p-value 0.0001, UKML 0.913 ppb yr$^{-1}$ with p-value < 0.0001, CAMS 0.569 ppb yr$^{-1}$ with p-value 0.0011, GEOS-Chem 0.164 ppb yr$^{-1}$ with p-value 0.5334, TCR-2 0.4 ppb yr$^{-1}$ with p-value 0.0343.**



**Figure 2: Population weighted ozone (OSMDA8) trends per decade for six datasets, calculated over the full period analyzed for each dataset. The different regions are defined in Table S7. Population weighted ozone (OSMDA8) trends per decade for six datasets from 2006 to 2016 is shown in Figure S4.**





**Figure 3: For six datasets from 2006 to 2016, (a) the 11-year ensemble mean, and (b) the average of annual standard deviations. Ozone data are reported as OSMDA8. The mean and standard deviation for each year are shown in Figures S5 and S6.**



**Figure 4: The difference of OSDMA8 in each grid cell between the 11-year (2006-2016) mean of each of six datasets and the ensemble mean (Figure 3). Positive values indicate that the average estimate of the dataset is higher than the ensemble mean. Negative values indicate that the average estimate of the dataset is lower than the ensemble mean of the six datasets. Difference maps for each year are shown in Fig. S7.**






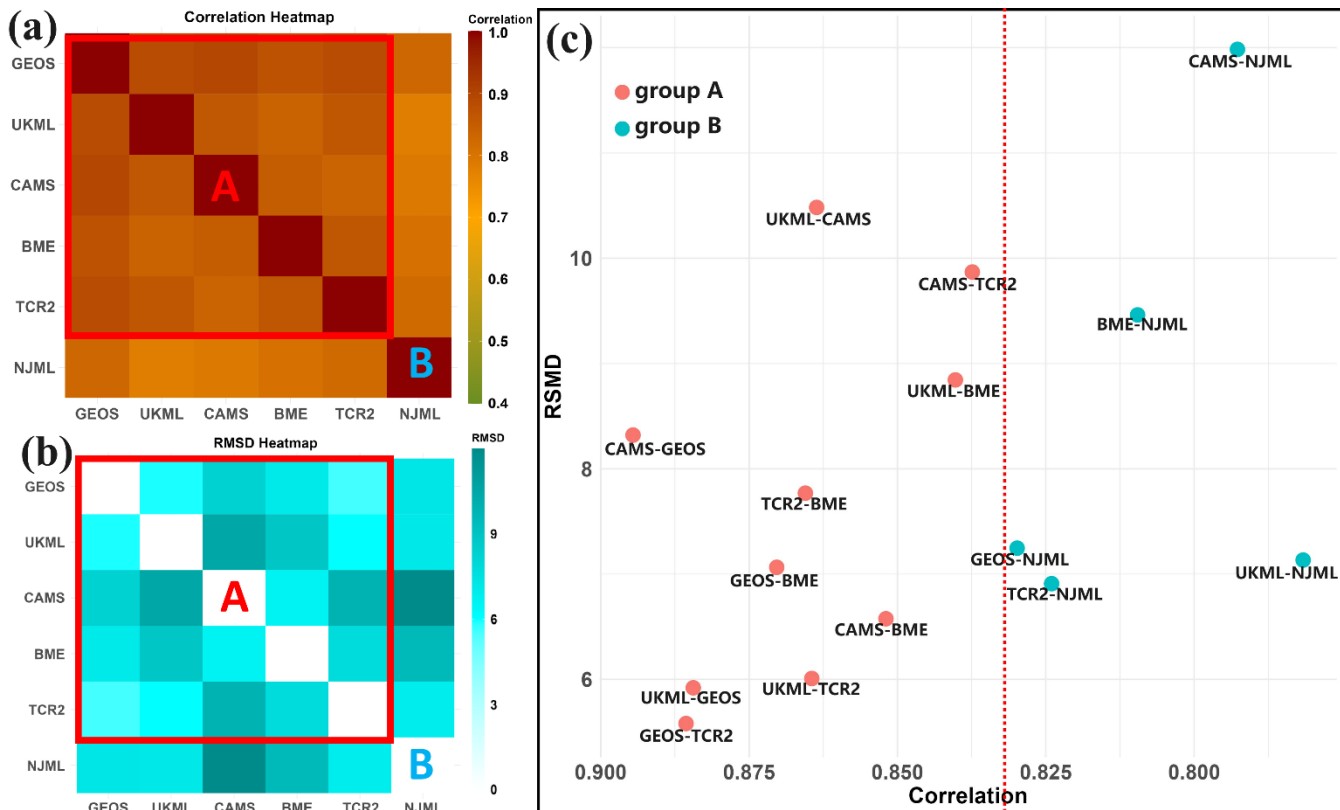

**Figure 5: Heatmaps of similarity among the six datasets, including (a) heatmaps of average of pairwise correlation (Pearson R) between each dataset from 2006 to 2016. (b) heatmaps of average of pairwise Root mean square difference (RMSD) between each dataset from 2006 to 2016. Group A designates five datasets with strong similarity, while Group B is composed of one dataset with lower similarity with the rest. (c) Scatterplot of correlation and RMSD between each pair of datasets. The datasets with greatest similarity are in the lower left of panel c, and comparisons with the Group B dataset have lower correlation. Heatmaps for each year are shown in Figure S8 and Figure S9.**



**Figure 6: Population exposed to 11-year average ozone (OSMDA8) from 2006 to 2016 in different regions. The horizontal axis represents ozone concentrations, and the vertical axis represents population size. Concentrations and population for each year are presented in Figure S10. The definitions of different regions are included in Table S7.**



**Table 2: The share of population exposed to ozone above particular thresholds in each world region, for the 2006 to 2016 average OSDMA8 for six ozone datasets. 0% means greater than 0 but less than 0.5%, 0 means no population share greater than this ozone concentration. Population shares for each year are shown in Table S8. The definitions of different regions are included in Table S7.**

| Dataset | BME | | | NJML | | | UKML | | | CAMS | | | GEOS | | | TCR-2 | | |
|---|---|---|---|---|---|---|---|---|---|---|---|---|---|---|---|---|---|---|
| Region | >30 | >50 | >70 | >30 | >50 | >70 | >30 | >50 | >70 | >30 | >50 | >70 | >30 | >50 | >70 | >30 | >50 | >70 |
| EAS | 100% | 61% | 0% | 100% | 72% | 3% | 100% | 99% | 31% | 100% | 67% | 0% | 100% | 95% | 4% | 100% | 94% | 41% |
| EUR | 99% | 16% | 0 | 100% | 76% | 0 | 99% | 77% | 0 | 98% | 9% | 0 | 100% | 44% | 0 | 100% | 70% | 0% |
| MDE | 100% | 79% | 0 | 100% | 99% | 5% | 99% | 94% | 0 | 100% | 88% | 8% | 100% | 99% | 4% | 100% | 94% | 38% |
| NAM | 99% | 17% | 0% | 100% | 88% | 3% | 99% | 84% | 0 | 100% | 40% | 0 | 100% | 55% | 0 | 99% | 86% | 0 |
| SAF | 93% | 3% | 0 | 99% | 36% | 0% | 98% | 10% | 0 | 86% | 8% | 0% | 99% | 14% | 0 | 98% | 18% | 1% |
| SAS | 100% | 89% | 0% | 100% | 99% | 8% | 100% | 99% | 40% | 99% | 96% | 12% | 99% | 95% | 0% | 99% | 90% | 10% |
| SEA | 84% | 0% | 0 | 89% | 27% | 0 | 97% | 41% | 0 | 88% | 24% | 6% | 89% | 0% | 0 | 85% | 13% | 0 |
| GLO | 96% | 42% | 0% | 99% | 70% | 4% | 98% | 69% | 16% | 93% | 48% | 4% | 98% | 59% | 1% | 97% | 64% | 13% |





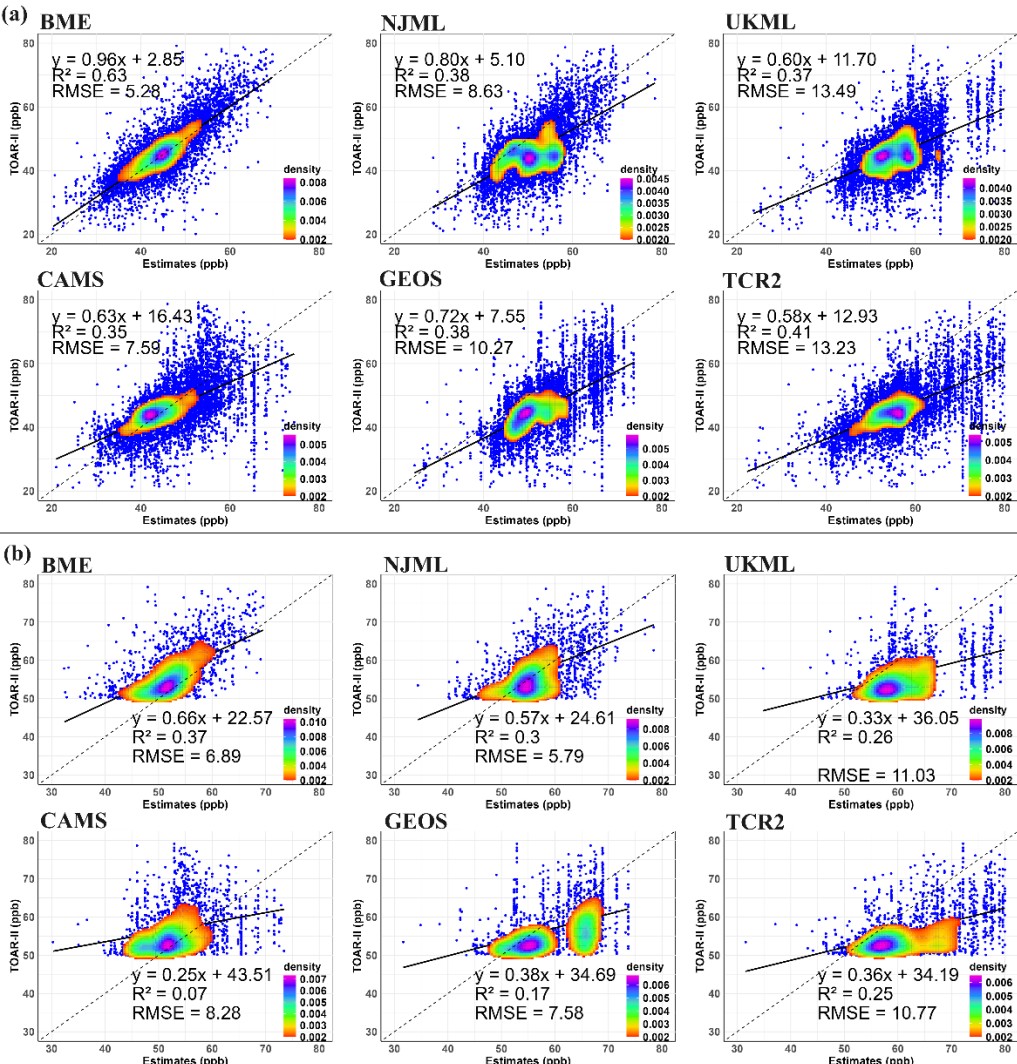

**Figure 7: Performance evaluations of six datasets with TOAR-II observations in 2016 for OSDMA8. The observation-prediction evaluations are presented in scatter plots with densities estimated by a Gaussian kernel function. Determination ($R^2$) and root mean**
**squared error (RMSE) are given. (a) The evaluation includes all monitor stations in the TOAR-II network in 2016. (b) The evaluation includes only monitor stations with observations above 50 ppb in the TOAR-II network in 2016. The dashed line marks where TOAR-II observations equal estimates (y=x line), and the solid black line represents the best-fit line. Performance evaluations for each year are shown in Figure S11 and Figure S12.**





**Figure 8: Normalized mean bias of six databases against TOAR-II observations (OSDMA8) at different quantiles in 2016. 0%: 13.46 ppb; 10%: 36.75 ppb; 20%: 39.80 ppb; 30%: 41.89 ppb; 40%: 43.57 ppb; 50%: 45.06 ppb; 60%: 46.82 ppb; 70%: 48.93 ppb; 80%: 52.18 ppb; 90%: 57.21 ppb; 100%: 86.25 ppb. Normalized mean bias for each year against TOAR-II observations are shown in Figure S14. Different quantiles of TOAR-II observations for other years are shown in Table S9.**





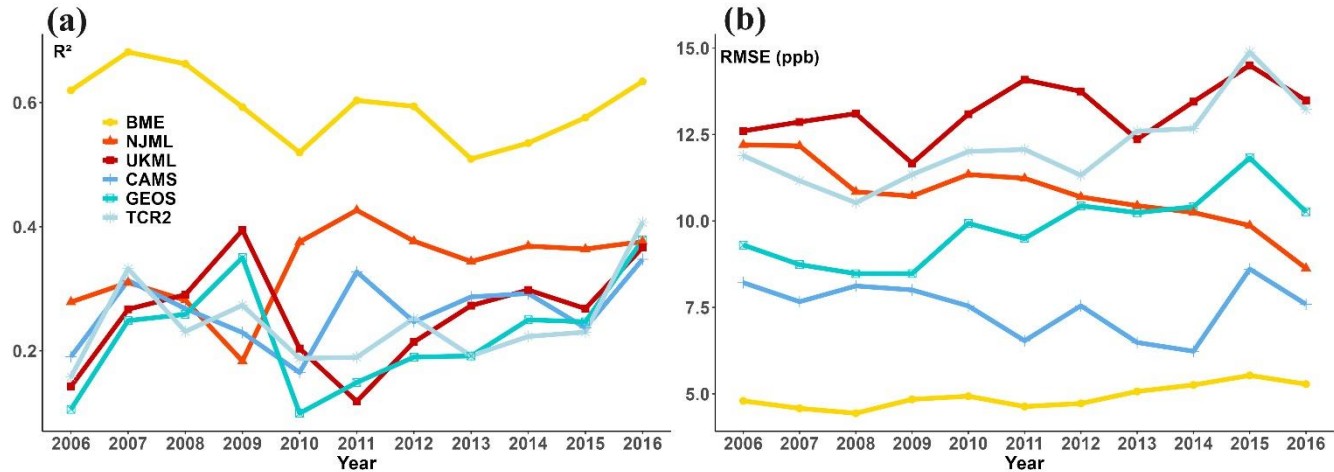

**Figure 9: (a) Time series of determination (R2) between each dataset and TOAR-II observations of OSDMA8 from 2006 to 2016. (b) Time series of root mean squared error (RMSE) between each dataset and TOAR-II from 2006 to 2016.**




**Table 3: Performance evaluation of six datasets for countries (union) with the most monitors in 2016 against TOAR-II observations of OSDMA8. Number is the number of the TOAR-II monitor stations in each country. Density (per km$^2$) is the density of the TOAR-II monitors in each country based on land area. Estimate is the average of the grid estimates for each dataset at the TOAR-II monitor locations in each country. Linear regression R$^2$ and root mean squared error (RMSE) against TOAR-II observations in each country are presented. Country names are United States of America (USA), China (CHN), Japan (JPN), South Korea (KOR), Canada (CAN). EU-27 includes Austria, Belgium, Bulgaria, Croatia, Cyprus, Czech Republic, Denmark, Estonia, Finland, France, Germany, Greece, Hungary, Ireland, Italy, Latvia, Lithuania, Luxembourg, Malta, Netherlands, Poland, Portugal, Romania, Slovakia, Slovenia, Spain, Sweden. Others is all other countries in TOAR-II apart from those listed. Performance evaluations for other years in these countries, are shown in Table S10.**

| Dataset | Country | EU-27 | USA | CHN | JPN | KOR | CAN | Others |
|---------|---------|-------|-----|-----|-----|-----|-----|--------|
| | Number | 2170 | 1425 | 1405 | 1108 | 315 | 260 | 330 |
| | Density | 5.43E-4 | 1.56E-4 | 1.50E-4 | 3.04E-3 | 3.23E-3 | 2.96E-5 | 1.07E-5 |
| | TOAR | 43.21 | 47.03 | 53.10 | 43.84 | 51.50 | 37.39 | 40.55 |
| BME | Estimate | 43.30 | 45.12 | 50.26 | 44.69 | 51.79 | 35.26 | 39.05 |
| | R$^2$ | 0.63 | 0.71 | 0.63 | 0.03 | 0.10 | 0.46 | 0.48 |
| | RMSE | 3.91 | 4.12 | 6.97 | 4.59 | 7.33 | 4.39 | 8.66 |
| NJML | Estimate | 53.53 | 48.44 | 53.39 | 49.40 | 54.62 | 43.79 | 48.63 |
| | R$^2$ | 0.41 | 0.58 | 0.57 | 0.00 | 0.07 | 0.30 | 0.51 |
| | RMSE | 11.49 | 4.55 | 6.86 | 7.42 | 8.01 | 7.57 | 11.59 |
| UKML | Estimate | 53.27 | 52.54 | 66.78 | 61.45 | 65.02 | 46.87 | 49.01 |
| | R$^2$ | 0.21 | 0.38 | 0.37 | 0.01 | 0.01 | 0.33 | 0.32 |
| | RMSE | 11.54 | 7.52 | 16.40 | 18.25 | 15.54 | 10.32 | 13.01 |
| CAMS | Estimate | 42.17 | 49.67 | 53.85 | 44.91 | 58.93 | 39.54 | 39.84 |
| | R$^2$ | 0.32 | 0.34 | 0.07 | 0.01 | 0.01 | 0.28 | 0.39 |
| | RMSE | 5.75 | 6.65 | 10.62 | 4.95 | 12.46 | 4.63 | 9.40 |
| GEOS | Estimate | 49.76 | 50.58 | 60.48 | 56.99 | 65.94 | 45.73 | 44.54 |
| | R$^2$ | 0.30 | 0.39 | 0.37 | 0.03 | 0.00 | 0.44 | 0.31 |
| | RMSE | 8.41 | 6.08 | 11.15 | 13.94 | 16.36 | 9.08 | 10.70 |
| TCR-2 | Estimate | 51.83 | 55.54 | 66.43 | 58.37 | 67.87 | 45.97 | 48.32 |
| | R$^2$ | 0.33 | 0.23 | 0.36 | 0.00 | 0.02 | 0.43 | 0.54 |
| | RMSE | 10.15 | 10.58 | 15.99 | 16.69 | 18.53 | 9.84 | 11.43 |



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
