# Peer review of "Intercomparison of global ground-level ozone datasets for healthrelevant metrics"

_EGUsphere, 2024_

## Referee Comment (RC1)

Review of the manuscript of egusphere-2024-3723 titled "Intercomparison of global ground-level ozone datasets for health-relevant metrics" written by Wang et al.

This study conducted a variety of analyses, including assessing ozone-exposure populations using extensive reanalysis and AI-derived ozone concentration data. While the analysis method itself is not entirely novel, the study is meaningful in its comparison of AI-based data with chemical reanalysis data. However, the authors have some issues that require improvement in the manuscript for publication. The following are the reviewer's concerns:

**Major comments**

1. Correct trend calculation and null hypothesis: Trends can vary depending on the selected time range. For instance, as shown in the figure below (Fig. R1), when restricting to the time range of GEOS data, the trends of BME and CAMS seem to be stagnant or declined, unlike those described in the manuscript. Consequently, if this time range is not properly justified, the calculated one itself may be questionable. Therefore, I strongly recommend that the authors provide a clear reason for selecting the different time ranges used in the trend calculation and assess its statistical significance.

[Figure]

Fig. R1. Six trends of OSMDA8 modified from Figure 1 in the manuscript.

2. Figure 2: Regarding the first comment, the comparison among the six datasets in Figure 2 (and discussion in Section 4.1) is meaningless since their temporal ranges are different.

3. Impact of data uncertainty on related analysis and reorganizing structure: The accuracy of predicted $O_3$ concentrations in each dataset significantly affects trend analysis, spatial distribution, and assessments of ozone-exposed populations. The substantial differences in uncertainty among the predicted datasets, as demonstrated through the comparison between TOAR-II observations and various predicted datasets in Figure 7, significantly hamper trend analysis and understanding of the ozone-exposure population. However, this study does not reflect or discuss the uncertainty in the several analyses presented in Section 4. Therefore, I recommend that the authors explicitly address the impact of dataset uncertainty on trend analysis and ozone-exposed population assessments. In addition, to discuss this efficiently, Section 4 and Section 5 should be re-arranged.

4. Also, regarding the third comment, one idea might be to compare the population exposure to ozone (i.e., Figure 6) calculated based on observations and six analysis datasets for the ozone observational (TOAR-II) sites.

5-1. Figure 7b. Why is the standard set at 50 ppb? What are the intended messages from the analysis in Figure 7b?

5-2. Fig. 7b (and Figure 8). If it is significant that the accuracy of prediction is lowered, particularly over 50 ppb, then how should the results in Figure 7b (or Figure 8) be considered in the analysis of Figure 6? It is also regarding the third comment.

6. Sect. 3.3 (Lines 224-225). For this case mentioned in lines 224-225, the observation data lack representativeness due to the coarse grid resolution in the GEOS-CHEM, CAMS, and TCR-2 datasets. Thus, the authors need to justify it.

7. L283 - 294. I would like to ask the authors to describe the purpose of separating Groups A and B in Figure 5. Additionally, please specify the criteria used to assign NJML to Group B. If the criterion is a correlation of ~0.83, what is the rationale behind this choice? Why was the RSMD criterion deemed unsuitable? Considering the statement in lines 289-290, the criteria appear to be arbitrary.

8. L328-338. Some statements lack objective descriptions based on consistent criteria. For example, it is stated that the TCR-2 shows adequate performance, whereas UKML has a significant overestimation. However, both datasets demonstrate similar performance in terms of correlation, RMSE, and slope for each year (refer to the tables below, with values taken from Figures 7 and S11). In fact, the lower slope in TCR-2 indicates a greater overestimation, so the description needs to be corrected.

| R² | BME | NJ | UK | CAMS | GEOS | TCR2 |
|---|---|---|---|---|---|---|
| 2006 | 0.62 | 0.28 | 0.14 | 0.19 | 0.11 | 0.16 |
| 2007 | 0.68 | 0.31 | 0.27 | 0.31 | 0.25 | 0.33 |
| 2008 | 0.66 | 0.28 | 0.29 | 0.27 | 0.26 | 0.23 |
| 2009 | 0.59 | 0.18 | 0.39 | 0.23 | 0.35 | 0.27 |
| 2010 | 0.52 | 0.38 | 0.2 | 0.17 | 0.1 | 0.19 |
| 2011 | 0.6 | 0.43 | 0.12 | 0.33 | 0.15 | 0.19 |
| 2012 | 0.59 | 0.38 | 0.21 | 0.25 | 0.19 | 0.25 |
| 2013 | 0.51 | 0.34 | 0.27 | 0.29 | 0.19 | 0.19 |
| 2014 | 0.53 | 0.37 | 0.3 | 0.29 | 0.25 | 0.22 |
| 2015 | 0.58 | 0.36 | 0.27 | 0.24 | 0.25 | 0.23 |
| 2016 | 0.63 | 0.38 | 0.37 | 0.35 | 0.38 | 0.41 |

| RMSE | BME | NJ | UK | CAMS | GEOS | TCR2 |
|---|---|---|---|---|---|---|
| 2006 | 4.8 | 12.2 | 12.6 | 8.21 | 9.3 | 11.89 |
| 2007 | 4.58 | 12.17 | 12.86 | 7.66 | 8.74 | 11.16 |
| 2008 | 4.44 | 10.84 | 13.1 | 8.12 | 8.48 | 10.53 |
| 2009 | 4.84 | 10.72 | 11.67 | 8 | 8.48 | 11.34 |
| 2010 | 4.93 | 11.34 | 13.09 | 7.54 | 9.93 | 12.01 |
| 2011 | 4.63 | 11.23 | 14.08 | 6.53 | 9.49 | 12.07 |
| 2012 | 4.72 | 10.69 | 13.75 | 7.55 | 10.44 | 11.32 |
| 2013 | 5.07 | 10.44 | 12.36 | 6.48 | 10.24 | 12.59 |
| 2014 | 5.26 | 10.24 | 13.45 | 6.23 | 10.41 | 12.67 |
| 2015 | 5.53 | 9.87 | 14.5 | 8.61 | 11.82 | 14.88 |
| 2016 | 5.28 | 8.63 | 13.49 | 7.59 | 10.27 | 13.23 |

| Slope | BME | NJ | UK | CAMS | GEOS | TCR2 |
|---|---|---|---|---|---|---|
| 2006 | 0.94 | 0.54 | 0.49 | 0.45 | 0.48 | 0.46 |
| 2007 | 0.97 | 0.61 | 0.68 | 0.57 | 0.66 | 0.68 |
| 2008 | 0.94 | 0.62 | 0.56 | 0.66 | 0.64 | 0.52 |
| 2009 | 0.89 | 0.52 | 0.74 | 0.46 | 0.7 | 0.59 |
| 2010 | 0.8 | 0.62 | 0.53 | 0.47 | 0.4 | 0.41 |
| 2011 | 0.91 | 0.65 | 0.37 | 0.6 | 0.53 | 0.48 |
| 2012 | 0.89 | 0.69 | 0.52 | 0.65 | 0.52 | 0.55 |
| 2013 | 0.79 | 0.68 | 0.56 | 0.57 | 0.47 | 0.4 |
| 2014 | 0.8 | 0.73 | 0.52 | 0.6 | 0.54 | 0.43 |
| 2015 | 0.93 | 0.75 | 0.49 | 0.51 | 0.54 | 0.42 |
| 2016 | 0.96 | 0.80 | 0.6 | 0.63 | 0.72 | 0.58 |

L329: I disagree with the characterization of the decreased as "minor". The $R^2$ value decreased significantly, from 0.63 to 0.51, which cannot be considered minor.

L330: The phrase "relatively good" is inappropriate. The performance is not good. It is better described as moderate.

**Minor comments**

1. Tables 1 – 6 are not mentioned in the manuscript. The authors need to check the order and ensure proper mention of all tables and figures.

2. L108: Provide an explanation of what M3Fusion is.

3. OSDMA8 and OSMDA8: These terms are used interchangeably. Check if it is correct, and if not, check the spelling.

4. In Section 4.1: Clarify what "area-weighted" and "population-weighted" mean or describe how they are calculated. Regarding this in Fig. 1, explain why the population-weighted mean increases more rapidly than the area-weighted one.

5. Y-axis in Figure 1: To avoid confusion, make the y-axis the same.

6. L269: Modify the phrase to "in the multi-model average over 50 ppb" in Line 269. Remove a dot before the 'over'.

7. Figures 7 and S11 – S13: The observation-prediction data points are shown in blue, which can be confused as indicating density. Thus, it would be better to change their color to black or gray for clarity.

8. Colors in Figures 1 and S3 (and Figures 8 and S14): To reduce confusion, use consistent color for each dataset across the figures.

9. L325: It seems that Figure S7 is mistakenly referenced and should be corrected to Figure S11.

10. Significant digits in Figures 7 and S11 – S12: Ensure that significant digits are presented consistently.

---

## Community Comment (CC1)

February 12, 2025

Comments by Owen R. Cooper (TOAR Scientific Coordinator of the Community Special Issue) on:

**Intercomparison of global ground-level ozone datasets for health-relevant metrics**

Hantao Wang, Kazuyuki Miyazaki, Haitong Zhe Sun, Zhen Qu, Xiang Liu, Antje Inness, Martin Schultz, Sabine Schröder, Marc Serre, and J. Jason West

EGUsphere [preprint], https://doi.org/10.5194/egusphere-2024-3723
Discussion started Jan. 3, 2025
Discussion closes Feb. 14, 2025

This review is by Owen Cooper, TOAR Scientific Coordinator of the TOAR-II Community Special Issue. I, or a member of the TOAR-II Steering Committee, will post comments on all papers submitted to the TOAR-II Community Special Issue, which is an inter-journal special issue accommodating submissions to six Copernicus journals:  ACP (lead journal), AMT, GMD, ESSD, ASCMO and BG. The primary purpose of these reviews is to identify any discrepancies across the TOAR-II submissions, and to allow the author teams time to address the discrepancies.  Additional comments may be included with the reviews. While O. Cooper and members of the TOAR Steering Committee may post open comments on papers submitted to the TOAR-II Community Special Issue, they are not involved with the decision to accept or reject a paper for publication, which is entirely handled by the journal's editorial team.

**Comments regarding TOAR-II guidelines:**

TOAR-II has produced two guidance documents to help authors develop their manuscripts so that results can be consistently compared across the wide range of studies that will be written for the TOAR-II Community Special Issue.  Both guidance documents can be found on the TOAR-II webpage: https://igacproject.org/activities/TOAR/TOAR-II

*The TOAR-II Community Special Issue Guidelines*:  In the spirit of collaboration and to allow TOAR-II findings to be directly comparable across publications, the TOAR-II Steering Committee has issued this set of guidelines regarding style, units, plotting scales, regional and tropospheric column comparisons, and tropopause definitions.

*The TOAR-II Recommendations for Statistical Analyses*:  The aim of this guidance note is to provide recommendations on best statistical practices and to ensure consistent communication of statistical analysis and associated uncertainty across TOAR publications. The scope includes approaches for reporting trends, a discussion of strengths and weaknesses of commonly used techniques, and calibrated language for the communication of uncertainty. Table 3 of the TOAR-II statistical guidelines provides calibrated language for describing trends and uncertainty, similar to the approach of IPCC, which allows trends to be discussed without having to use the problematic expression, "statistically significant".

**General comments:**

Line 23
Is there any reason to report 60.8% with one decimal place? Would 61% be better, given the uncertainty in the estimate?

Line 24
The following statement is not very clear:
"These differences are large enough to impact health and other applications."
I suggest:
"These differences are large enough to impact assessments of health impacts and other applications."

Line 34
Please also provide the uncertainty range, along with the estimate of mortality.

Line 38
Make it clear that these ozone increases refer to ozone above the surface (surface ozone was not reported in this study because the surface observations were from airport runways, which are not representative of typical conditions). When mentioning population-weighted metrics, Gaudel et al. (2020) is not a correct reference as it does not address these metrics. Please provide a different reference. It would be helpful to list some references that provide recent updates on surface ozone trends. One such paper is Chang et al. (2024), submitted to the TOAR-II special issue, which focuses on long-term surface ozone trends across the USA.

Line 243-244
It is an oversimplification to say that ozone is typically increasing in the northern hemisphere over 2005-2016. First you need to specifically state that you are talking about the OSDMA8 metric, which is very different from the metrics reported by Gaudel et al (2018) and Fleming et al. (2018). These earlier studies showed a range of increasing and decreasing ozone trends that varied by region. The recent trend update by Chang et al. (2024) shows decreasing ozone in the eastern and western USA over the period 2005-2016. I recommend that you refer to studies that have focused on OSDMA8, such as Becker et al., 2023, and Malashock et al., 2022 (see Figure 1 and Figure 2 of Malashock et al., 2022; note that this is the second paper by Malashock, published in 2022; see the reference listed below).

Line 509
According to the TOAR data use policy (https://toar-data.fz-juelich.de/footer/terms-of-use.html), the TOAR data also needs the following citation:
Schröder et al; TOAR Data Infrastructure;
https://doi.org/10.34730/4d9a287dec0b42f1aa6d244de8f19eb3

Figure 1
Following the TOAR-II statistical guidelines, all trends need to be reported with their 95% confidence intervals and *p*-values.

Figure 7
These figures need to be reoriented, with the TOAR-II observation being the independent variable on the x-axis, and the model output being the dependent variable on the y-axis.

**References**

Chang, K.-L., McDonald, B. C., and Cooper, O. R. (2024), Surface ozone trend variability across the United States and the impact of heatwaves (1990–2023), EGUsphere [preprint], https://doi.org/10.5194/egusphere-2024-3674 (submitted to ACP as a contribution to the TOAR-II Community Special Issue)

Malashock, Daniel A., Marissa N. Delang, Jacob S. Becker, Marc L. Serre, J. Jason West, Kai-Lan Chang, Owen R. Cooper, Susan C. Anenberg (2022), Global Trends in Ozone Concentration and Attributable Mortality for Urban, Peri-Urban and Rural Areas between 2000 and 2019: A Modelling Study, The Lancet Planetary Health, Volume 6, Issue 12, Pages E958-E967, https://doi.org/10.1016/S2542-5196(22)00260-1

---

## Author Comment (AC1)

Review of the manuscript of egusphere-2024-3723 titled "Intercomparison of global ground-level ozone datasets for health-relevant metrics" written by Wang et al.

This study conducted a variety of analyses, including assessing ozone-exposure populations using extensive reanalysis and AI-derived ozone concentration data. While the analysis method itself is not entirely novel, the study is meaningful in its comparison of AI-based data with chemical reanalysis data. However, the authors have some issues that require improvement in the manuscript for publication. The following are the reviewer's concerns:

*Response:*
*We thank the reviewer for their careful reading of the manuscript and thoughtful comments.*

**Major comments**

1. Correct trend calculation and null hypothesis: Trends can vary depending on the selected time range. For instance, as shown in the figure below (Fig. R1), when restricting to the time range of GEOS data, the trends of BME and CAMS seem to be stagnant or declined, unlike those described in the manuscript. Consequently, if this time range is not properly justified, the calculated one itself may be questionable. Therefore, I strongly recommend that the authors provide a clear reason for selecting the different time ranges used in the trend calculation and assess its statistical significance.

[Figure]

Fig. R1. Six trends of OSMDA8 modified from Figure 1 in the manuscript.

*Response:*
*Thank you for this comment. We agree with your points that the selection of the time range can significantly influence trend calculations, and that choosing a uniform time range over all datasets gives the most consistent basis for comparison. We added a table (Table 2) that limits analysis of trends to 2006 to 2016, showing the area-weighted trend and population-weighted trend for six datasets with 95% UI in the main manuscript. We also maintain Figure 1 because it remains valuable for illustrating the trends over each dataset's full coverage period. And we add a table (Table S13) for the full time period with 95% UI in the SI. The text is revised in some places to discuss results when focusing on 2006-2016.*
*Revised:*

*Line 263: "In Table 2, focusing on the period from 2006 to 2016, we find that NJML is the only dataset showing a downward trend in both area-weighted and population-weighted mean*

*ozone concentrations, with very high certainty. In contrast, TCR-2 and UKML show increasing trends in population-weighted mean ozone during this period with very high certainty."*

*Line 415: "NJML demonstrates a decreasing trend in global population-weighted and area-weighted yearly mean over the 2006-2016 period, while the five others exhibit either increasing trends or no clear trend."*

*Line 498: " In terms of long-term trends over 2006 to 2016 period, UKML and TCR-2 show a consistent upward trend globally, while NJML shows a downward trend."*

2. Figure 2: Regarding the first comment, the comparison among the six datasets in Figure 2 (and discussion in Section 4.1) is meaningless since their temporal ranges are different.

*Response:*
*In the original draft, Figure S4 shows the regional trend from 2006 to 2016. We now move that figure to the main body to replace Figure 2 and move the original Figure 2 to Figure S4 with a note that time periods are inconsistent. We also add a table in SI (Table S11) to show the trend of six datasets in each region from 2006 to 2016, with 95% UI.*
*Revised:*
*Line 22: "For example, in Europe, the two chemical reanalyses show an increasing trend while the other datasets show no increase."*
*Line 275: "From Table S11, we observe that some regions exhibit a clearer trend from 2006 to 2016, with very high certainty across six datasets. In East Asia, BME and NJML observe decreasing trends, whereas the other 4 datasets display increasing trends. In North America, all datasets display a downward trend, and in Europe, BME, NJML, UKML and TCR-2 show a decline, contrasting with increases in CAMS and GEOS-chem. Recent analyses using TOAR observations indicate that from 2005 to 2016, most sites in North America experienced decreasing ozone, while many sites in East Asia exhibited significant positive trends."*
*Line 416: "Divergence among datasets becomes even more evident in the analysis of regional ozone trends (Fig. 2). The ozone concentrations decreased in Europe from 2006 to 2016 according to BME, NJML, UKML, and TCR-2, yet increase in the other chemical reanalysis datasets"*
*Line 429: "In Fig. 2, all datasets exhibit a downward trend in North America over 2006 to 2016."*
*Line 499: "Regionally, all datasets show a downward trend in North America, and only BME and NJML datasets demonstrate a downward trend in East Asia; In Europe, BME, UKML, NJML and TCR-2 report a downward trend, while the other two chemical reanalysis datasets reveal an upward trend that is not seen in observations."*

3. Impact of data uncertainty on related analysis and reorganizing structure: The accuracy of predicted $O_3$ concentrations in each dataset significantly affects trend analysis, spatial distribution, and assessments of ozone-exposed populations. The substantial differences in uncertainty among the predicted datasets, as demonstrated through the comparison between TOAR-II observations and various predicted datasets in Figure 7, significantly hamper trend analysis and understanding of the ozone-exposure population. However, this study does not reflect or discuss the uncertainty in the several analyses presented in Section 4. Therefore, I recommend that the authors explicitly address the impact of dataset uncertainty on trend analysis and ozone-exposed population assessments. In addition, to discuss this efficiently, Section 4 and Section 5 should be re-arranged.

*Response:*

*We appreciate your feedback. We agree that the analyses of trends, spatial distributions, and population exposure among the different datasets in Section 4 can be informed by comparisons of each dataset with observations in Section 5. The information in Section 5 provides some guidance as to which dataset in Section 4 is likely to be closer to reality. However, there is also logic in showing how each dataset compares with the others comprehensively, showing differences among datasets and for application to population exposure, before comparing datasets with observations. Therefore, we have chosen to keep the original organization in Sections 4 and 5. Readers who wish to can view the agreement with observations in Section 5 to make their own judgements about the likely veracity of the different datasets shown in Section 4. Then following Sections 4 and 5, causes of uncertainties in the datasets and their relevance for trend analysis and population exposure assessments are discussed in Section 6.*

*In fact, we attempted to use the biases identified in Section 5 to interpret and discuss the comparative results presented in Section 4; however, we did not find a clear and effective approach. Instead, in Section 6, we explicitly discuss how each model's overestimations and underestimations impact the differences observed in the comparative analyses of Section 4. And we add that uncertainties in each dataset impact the accuracy of trend analyses and population exposure assessments in Section 6.*

*Revised:*

*Line 454: "The performance of each dataset can impact the accuracy of trend analysis (Fig. 1 and Fig. 2) and population exposure assessment (Fig. 6), which may lead to very different results when compared to the WHO guideline and interim target."*

4. Also, regarding the third comment, one idea might be to compare the population exposure to ozone (i.e., Figure 6) calculated based on observations and six analysis datasets for the ozone observational (TOAR-II) sites.

*Response:*

*We appreciate the suggestion to compare population exposure based on observations with that derived from the six analysis datasets. However, directly calculating population ozone exposure from TOAR-II observations is subject to high uncertainty because the monitoring stations are sparsely distributed, and some method would be needed to interpolate between the observations, and this is similar to what the geostatistical and machine learning datasets do.*

*Additionally, our analysis focused on comparing the six datasets to understand their differences in section 4, and we have thoroughly evaluated the performance of each model against TOAR-II data across different concentrations, regions, and years in section 5.*

5-1. Figure 7b. Why is the standard set at 50 ppb? What are the intended messages from the analysis in Figure 7b?

**Response:**

*We selected 50 ppb because it corresponds to the long-term air quality interim target established by the WHO, as stated in the description we added in the Section 2 data part. Figure 7b is intended to demonstrate each dataset's capability to capture ozone concentrations exceeding this ozone level, highlighting their ability to detect years of high ozone. We have revised the discussion of Figure 7b in the second paragraph of Section 5.5 for better clarity.*

**Revised:**

*Line 108: "OSDMA8 is GBD's ozone metric for quantifying health effect from long-term ozone exposure (Brauer et al., 2024), and it is the metric used in the World Health Organization's air quality guidelines, with values of 30 ppb for the guideline and 50 ppb for the interim target (World Health Organization, 2021)."*

*Line 358: "Fig. 7(b) focuses only on TOAR-II sites with OSDMA8 value above 50 ppb, showing that $R^2$ is reduced compared to the comparison of all ozone measurements (Fig. 7(a)) for all six datasets, suggesting overall weaker agreement between modeled and observed ozone distributions at higher concentrations."*

*Line 361: "However, the change of biases varies among datasets at higher concentrations. Specifically, overestimation is reduced in the UKML, NJML, GEOS-Chem, and TCR-2 datasets when observations exceed 50 ppb. Conversely, we observe increased underestimation in the BME and CAMS datasets at ozone levels above 50 ppb."*

5-2. Fig. 7b (and Figure 8). If it is significant that the accuracy of prediction is lowered, particularly over 50 ppb, then how should the results in Figure 7b (or Figure 8) be considered in the analysis of Figure 6? It is also regarding the third comment.

**Response:**

*Yes, we agree with your point that model's accuracy varies at high ozone levels. The comparison in Figure 6 is mainly to address the fact that researchers would typically use any of the six models as the basis for health-related studies on ozone concentrations. We should take into account differences in exposure estimates among the datasets without recalibrating or correcting them.*

*We have clarified our discussion of Figures 7 and 8 by explicitly noting that the poorer performance of some datasets at higher ozone concentrations will influence the distribution of ozone exposure across the population, as presented in Figure 6.*

**Revised:**

*Line 454: "The performance of each dataset can impact the accuracy of trend analysis (Fig. 1 and Fig. 2) and population exposure assessment (Fig. 6), which may lead to very different results when compared to the WHO guideline and interim target."*

6. Sect. 3.3 (Lines 224-225). For this case mentioned in lines 224-225, the observation data lack representativeness due to the coarse grid resolution in the GEOS-CHEM, CAMS, and TCR-2 datasets. Thus, the authors need to justify it.

*Response:*

*For grid cells with TOAR-II observations, the GEOS-CHEM, CAMS, and TCR-2 reanalysis datasets did not have any missing values. Only the BME, NJML and UKML dataset exhibited some NA values (at finer resolutions). We add a table (S12) detailing the number of NA values and sample sizes for each dataset in SI.*

*Original text:*

*For grid cells with a TOAR-II observation but no valid estimate in a dataset (NA value), we used the nearest valid estimate instead.*

*Revised:*

*Line 244: "For grid cells with a TOAR-II observation but no valid estimate in a dataset (NA value), we used the nearest valid estimate instead. Table S12 displays the number of missing values in each dataset in 2016 at TOAR-II locations, showing that only BME, NJML and UKML have a small number of missing estimates."*

7. L283 - 294. I would like to ask the authors to describe the purpose of separating Groups A and B in Figure 5. Additionally, please specify the criteria used to assign NJML to Group B. If the criterion is a correlation of ~0.83, what is the rationale behind this choice? Why was the RSMD criterion deemed unsuitable? Considering the statement in lines 289-290, the criteria appear to be arbitrary.

*Response:*

*We separated the datasets into Groups A and B to compare their spatial distribution patterns of ozone estimates. Our grouping method is based on pairwise correlation rather than RMSD because our focus is on spatial similarity, not absolute magnitude differences. Although a correlation value around 0.83 is mentioned, it is not used as a strict criterion. The objective is to ascertain the grouping combination that maximizes the difference between the mean of the within-group correlations and the mean of the out-of-group correlations. The details of the grouping method are described in Text S1. Moreover, even though datasets in Group A show similar spatial distributions, the high RMSD values among them reveal significant differences in the ozone estimates. We add more descriptions of this grouping method in the main manuscript.*

*Revised:*

*Line 232: "The idea of this grouping is to distinguish the spatial similarity between the datasets, which is based on the pairwise correlation. For each grouping combination, 4 variables are computed: the sum of pairwise correlations within groups ($C_i$), the sum of pairwise correlations outside the groups ($C_o$), the number of dataset pairs within groups ($N_i$), and the number of dataset pairs outside the groups ($N_o$). The objective is to ascertain the grouping combination that maximizes the difference between $C_i/N_i$ and $C_o/N_o$. More details of the calculation can be found in Text S1."*

8. L328-338. Some statements lack objective descriptions based on consistent criteria. For example, it is stated that the TCR-2 shows adequate performance, whereas UKML has a significant overestimation. However, both datasets demonstrate similar performance in terms of correlation, RMSE, and slope for each year (refer to the tables below, with values taken from Figures 7 and S11). In fact, the lower slope in TCR-2 indicates a greater overestimation, so the description needs to be corrected.

| $R^2$ | BME | NJ | UK | CAMS | GEOS | TCR2 |
|---|---|---|---|---|---|---|
| 2006 | 0.62 | 0.28 | 0.14 | 0.19 | 0.11 | 0.16 |
| 2007 | 0.68 | 0.31 | 0.27 | 0.31 | 0.25 | 0.33 |
| 2008 | 0.66 | 0.28 | 0.29 | 0.27 | 0.26 | 0.23 |
| 2009 | 0.59 | 0.18 | 0.39 | 0.23 | 0.35 | 0.27 |
| 2010 | 0.52 | 0.38 | 0.2 | 0.17 | 0.1 | 0.19 |
| 2011 | 0.6 | 0.43 | 0.12 | 0.33 | 0.15 | 0.19 |

| | BME | NJ | UK | CAMS | GEOS | TCR2 |
|---|---|---|---|---|---|---|
| 2012 | 0.59 | 0.38 | 0.21 | 0.25 | 0.19 | 0.25 |
| 2013 | 0.51 | 0.34 | 0.27 | 0.29 | 0.19 | 0.19 |
| 2014 | 0.53 | 0.37 | 0.3 | 0.29 | 0.25 | 0.22 |
| 2015 | 0.58 | 0.36 | 0.27 | 0.24 | 0.25 | 0.23 |
| 2016 | 0.63 | 0.38 | 0.37 | 0.35 | 0.38 | 0.41 |
| **Slope** | **BME** | **NJ** | **UK** | **CAMS** | **GEOS** | **TCR2** |
| 2006 | 0.94 | 0.54 | 0.49 | 0.45 | 0.48 | 0.46 |
| 2007 | 0.97 | 0.61 | 0.68 | 0.57 | 0.66 | 0.68 |
| 2008 | 0.94 | 0.62 | 0.56 | 0.66 | 0.64 | 0.52 |
| 2009 | 0.89 | 0.52 | 0.74 | 0.46 | 0.7 | 0.59 |
| 2010 | 0.8 | 0.62 | 0.53 | 0.47 | 0.4 | 0.41 |
| 2011 | 0.91 | 0.65 | 0.37 | 0.6 | 0.53 | 0.48 |
| 2012 | 0.89 | 0.69 | 0.52 | 0.65 | 0.52 | 0.55 |
| 2013 | 0.79 | 0.68 | 0.56 | 0.57 | 0.47 | 0.4 |
| 2014 | 0.8 | 0.73 | 0.52 | 0.6 | 0.54 | 0.43 |
| 2015 | 0.93 | 0.75 | 0.49 | 0.51 | 0.54 | 0.42 |
| 2016 | 0.96 | 0.80 | 0.6 | 0.63 | 0.72 | 0.58 |

| **RMSE** | **BME** | **NJ** | **UK** | **CAMS** | **GEOS** | **TCR2** |
|---|---|---|---|---|---|---|
| 2006 | 4.8 | 12.2 | 12.6 | 8.21 | 9.3 | 11.89 |
| 2007 | 4.58 | 12.17 | 12.86 | 7.66 | 8.74 | 11.16 |
| 2008 | 4.44 | 10.84 | 13.1 | 8.12 | 8.48 | 10.53 |
| 2009 | 4.84 | 10.72 | 11.67 | 8 | 8.48 | 11.34 |
| 2010 | 4.93 | 11.34 | 13.09 | 7.54 | 9.93 | 12.01 |
| 2011 | 4.63 | 11.23 | 14.08 | 6.53 | 9.49 | 12.07 |
| 2012 | 4.72 | 10.69 | 13.75 | 7.55 | 10.44 | 11.32 |
| 2013 | 5.07 | 10.44 | 12.36 | 6.48 | 10.24 | 12.59 |
| 2014 | 5.26 | 10.24 | 13.45 | 6.23 | 10.41 | 12.67 |
| 2015 | 5.53 | 9.87 | 14.5 | 8.61 | 11.82 | 14.88 |
| 2016 | 5.28 | 8.63 | 13.49 | 7.59 | 10.27 | 13.23 |

*Response:*

*We acknowledge that our original description was misleading. In fact, TCR-2 indicates a greater overestimation compared to UKML. We revise the manuscript accordingly to provide a more objective description based on performance as shown in Figures 7 and S1.*

*Revised:*

*Line 353: "UKML exhibits the highest RMSE of 13.49 ppb, and its density core region is above the y=x dashed line, indicating an overestimation. This is because the UKML algorithm emphasizes higher ozone pollution levels in rural and remote areas compared to adjacent urban districts, which consequently leads to an overestimation especially in population-weighted metrics."*

9. L329: I disagree with the characterization of the decreased as "minor". The $R^2$ value decreased significantly, from 0.63 to 0.51, which cannot be considered minor.

*Response:*

*We change description to "significantly". After re-running the evaluation, the $R^2$ improves to 0.53. This time, we excluded all sites located at observation points previously used as BME input. In the initial version of the manuscript, we removed the nearest sites to the BME observations points if they were within a 1-degree radius. Compared to other datasets, 0.53 is still good performance.*

*Revised:*

*Line 344: "After excluding all sites located at observation points previously used as BME input, using 3911 observations for validation, BME performs well compared to another datasets, though its R2 decreases significantly to 0.53."*

10. L330: The phrase "relatively good" is inappropriate. The performance is not good. It is better described as moderate.

*Response:*

*We agree that "relatively good" overstates the performance. We revise the description to "moderate," which more accurately reflects the performance by the dataset.*

*Revised:*

*Line 346: "In Fig. 7(a), all three chemical reanalysis datasets exhibit a moderate $R^2$ ranging from 0.35 to 0.41, comparable to the performance of the machine learning datasets, which have $R^2$ values of 0.37 and 0.38."*

**Minor comments**

1. Tables 1 – 6 are not mentioned in the manuscript. The authors need to check the order and ensure proper mention of all tables and figures.

   ***Response:***
   *I add the Table numbers when I mention Tables S1-S6 in the main manuscript.*

2. L108: Provide an explanation of what M3Fusion is.

   ***Response:***

   *M3Fusion is a composite of multiple chemistry models by weighting based on their performance against TOAR observations.*

   ***Revised:***

   *Line 114: "M$^3$Fusion (Measurement and Multi-Model Fusion) is a statistical method developed to improve estimates of global surface ozone distributions by integrating observational data from TOAR and outputs from multiple chemistry models. Specifically, the method assigns weights to multiple atmospheric chemistry models based on their regional accuracy compared to observed ozone values."*

3. OSDMA8 and OSMDA8: These terms are used interchangeably. Check if it is correct, and if not, check the spelling.

   ***Response:***
   *We correct OSMDA8 to OSDMA8.*

4. In Section 4.1: Clarify what "area-weighted" and "population-weighted" mean or describe how they are calculated. Regarding this in Fig. 1, explain why the population-weighted mean increases more rapidly than the area-weighted one.
   ***Response:***
   *We add explain the potential reason that lead to rapidly increases in population-weigthed mean. We add the explanation of "area-weighted" and "population-weighted" in Text S2 in the SI with the calculation methods.*

   ***Revised:***

   *Line 216: "We calculated the yearly ozone trend for each dataset using both population-weighted and area-weighted approaches, with details of the calculation methods provided in Text S2."*

   *Line 263: "The faster increase in the population-weighted mean compared to the area-weighted mean appears to be driven by rising ozone levels in highly populated regions."*

5. Y-axis in Figure 1: To avoid confusion, make the y-axis the same.
   ***Response:***
   *We change the Y-axis to the same.*

6. L269: Modify the phrase to "in the multi-model average over 50 ppb" in Line 269. Remove a dot before the 'over'.

   ***Response:***
   *We remove the dot and revised the phrase.*

   ***Revised:***
   *Line 284: India, China, and the Middle East are estimated to have the world's highest average ozone concentrations, exceeding 50 ppb in the multi-model average.*

7. Figures 7 and S11 – S13: The observation-prediction data points are shown in blue,

which can be confused as indicating density. Thus, it would be better to change their color to black or gray for clarity.

*Response:*

*We used blue color to distinguish the y=x line from the regression line. We have changed the data points to grey color.*

8. Colors in Figures 1 and S3 (and Figures 8 and S14): To reduce confusion, use consistent color for each dataset across the figures.

*Response:*
*We have changed to use the same color.*

9. L325: It seems that Figure S7 is mistakenly referenced and should be corrected to Figure S11.

*Response:*
*We change it to S11.*

10. Significant digits in Figures 7 and S11 – S12: Ensure that significant digits are presented consistently.

*Response:*

*We changed the significant digits to be consistent for Figures 7 and S11 – S12.*

---

## Author Comment (AC2)

*We thank the reviewer for their attention to this manuscript and thoughtful comments.*

L23:  Given the large bias errors in these data sets, comparing the population exposed to a threshold value, like 50 ppb is meaningless.  Do these differences impact "health" as stated or one's analysis of health effects.  This is a bit sloppy writing.

***Response:***

*We selected the 50 ppb threshold because it corresponds to the long-term air quality interim target established by the WHO. Our analysis focused on comparing the six datasets to understand their differences. It is important to evaluate whether the large differences between ozone products result in different conclusions. We add more discussion in section 6 (Discussion), that biases in the datasets will affect assessments like the population exposure above 50 ppb. Then there is a need for future work that will reduce the bias of ozone products.*

***Revised:***
*Line 25: "These differences are large enough to impact assessments of health impacts and other applications."*

*Line 109: "OSDMA8 is GBD's ozone metric for quantifying health effect from long-term ozone exposure (Brauer et al., 2024), and it is the metric used in the World Health Organization's air quality guidelines, with values of 30 ppb for the guideline and 50 ppb for the interim target (World-Health-Organization, 2021)."*

*Line 251: "We selected the 50 ppb as the threshold for high ozone concentration because it corresponds to the long-term air quality interim target of WHO."*

*Line 454: "The performance of each dataset can impact the accuracy of trend analysis (Fig. 1 and Fig. 2) and population exposure assessment (Fig. 6), which may lead to very different results when compared to the WHO guideline and interim target."*

L25: very good point, but you should compare the gridded data you have with Schnell's gridding of the EU and NAm.  Comparing points to grid-cell averages that you have from the global data sets is a serious science problem -? not the way you treat it here.

How can you get an R2 for surface sites vs grid-cell means?? This is not sensible.

***Response:***

*Thank you for raising this important point. We acknowledge your concern regarding the comparison of grid-cell averages to point measurements. However, the evaluation approach we employed, using grid-cell averages of model output  to evaluate model performance at point locations of observations, is widely used for evaluation. The goal of*

*our work (Section 5) is to assess the accuracy of gridding products in estimating measured ground concentration point values.*

*Schnell et al.'s approach of creating 1º×1º grid-cell averages of TOAR observational data is valuable and effective for regions with dense monitoring networks, such as Europe and North America. However, there are several reasons to not use it in our study. First, we specifically focus on evaluating global ozone mapping products against the most recent TOAR-II observations. However, Schnell's dataset is inadequate to evaluate globally, and it was created using TOAR-I data. Second, most datasets included in our comparison have finer spatial resolutions than 1º×1º.*

*In this context, the grid-cell-average-to-station-point comparison represents an accepted method. We explicitly acknowledge the limitations of this approach in our manuscript and clarify that the performance metrics, including $R^2$ values, should be interpreted considering this spatial representativeness uncertainty.*

***Revised:***
*Line 252: "These performance metrics should be interpreted considering the spatial representativeness uncertainty that is caused by the grid-to-point evaluation approach."*

L27:  You just said your data is worst at overestimating O3 at low abundances, but here you say it is worse for >50 ppb??

***Response:***

*This decline in performance at higher ozone concentrations (>50 ppb) arises not primarily from increased overestimation but rather from reduced agreement between modeled and observed ozone distribution at the higher ozone concentration (>50 ppb).*

*Thus, there is no contradiction: the datasets typically overestimate ozone concentrations at lower observed levels, but the $R^2$ deteriorates more significantly at higher ozone concentrations due to increased uncertainty and reduced agreement at these more extreme conditions. We clarify this distinction more explicitly in the revised manuscript.*

***Revised:***

*Line 25: "Comparing with Tropospheric Ozone Assessment Report (TOAR) II ground-level observations, most datasets overestimate ozone, particularly at lower observed concentrations. In 2016, across all stations, $R^2$ ranges among the six datasets from 0.35 to 0.63, and RMSE from 5.28 to 13.49 ppb. Agreement between modeled and observed ozone distributions is reduced at ozone concentrations above 50 ppb."*

L29:  "highlighting the importance of continued research on global ozone distributions"

*Response:*

*The referee refers to the original text without comment. The large discrepancies found among datasets suggest that it is important to continue research on global ozone distributions through more widespread measurements, improved modeled estimates, etc. We retain the original text here.*

L38: Oh really. The number of regions could be much much greater if you picked smaller regions. The key issue is the area fraction NOT the number of regions.

*Response:*

*Thank you for highlighting the importance regarding area fraction. Gaudel et al. find that ozone is increasing over all 11 NH regions that they defined and analyzed, and they did not find decreasing or flat trends in any region.*

*Revised:*

*Line 40: "Gaudel et al. find that since the mid-1990s, tropospheric ozone above the surface has increased across all 11 study regions in the Northern Hemisphere that they defined and analyzed (Western North America, Eastern North America, Southeast North America, Northern South America, Northeast China/Korea, The Persian Gulf, India, Southeast Asia, Malaysia/Indonesia, Europe, Gulf of Guinea) (Gaudel et al., 2020)."*

L42: 30 ppb is basically the minimum background level – this is not a useful statement and it implies that pollution is the cause here.

*Response:*

*Thank you for this clarification. We show results for population exposure above 30 ppb because this is the WHO air quality guideline. We agree that this guideline is near the background ozone level, although typical estimates of preindustrial (without human influence) ozone are lower. The intention here was not to imply that ozone near 30 ppb results from pollution.*

L44: The quality of writing (logic, not English) is poor: You just quoted all these results that rely on estimates of surface ozone and then you say you lack knowledge of surface ozone.

*Response:*

*Thank you for pointing out the potential confusion. We recognize how the final sentence might seem logically inconsistent with the preceding context. Our intention was not to suggest that no knowledge exists regarding surface ozone concentrations; rather, we intended to highlight that despite existing assessments, substantial uncertainties remain*

*due to observational gaps, especially in remote and developing regions. The paragraph that follows this one focuses on recent research on ozone mapping products. We revise the last sentence of this paragraph to clarify this point explicitly.*

***Revised:***
*Line 48: "Despite existing assessments, substantial uncertainties remain due to observational gaps, especially in remote and developing regions."*

L70: Great. This is the most important statement. Could be up front

***Response:***

*Thank you for this comment. We agree that this statement is important, and we move it up to the end of second paragraph of the Introduction.*

L71: "Potential" – there are most assuredly inconsistencies.

***Response:***

*Thank you for highlighting this point. We acknowledge that inconsistencies among ozone datasets assuredly exist. Our original use of "potential" was too cautious. To reflect this clearly, we delete the "Potential".*

***Revise:***

*Line 77: "Inconsistencies in these datasets could significantly impact public health research, especially in assessing the risks of ozone-related health impacts, and may impede the development of effective environmental policies and ozone management strategies."*

L75: the biases and errors certainly come from the process. I hope you are not using 'data' to describe the assimilation modeling here.

***Response:***

*Thank you for highlighting this point. We fully agree that biases and errors primarily arise from the processes. In our original phrasing, "data" referred specifically to the "input data" utilized by each ozone mapping product. However, we recognize that the input data themselves can also contribute to these biases and errors. In the subsequent discussion, we explicitly note that chemical reanalysis products are constrained by limitations in satellite observations, while machine learning and geostatistical methods are constrained by the spatial distribution of ground-level monitoring stations.*

***Revised:***
*Line 79: "Although each dataset incorporates a considerable amount of observational*

*information and model simulations through various methodologies, each inherently incorporates biases from these input data sources during the fusion processes. "*

L85-95: This exposes the fundamental flaws in the focus of this paper. The use of OSDMA8 totally washes out the key fundamental information about ozone that can be tested with the real surface direct observations. The 24-hour diel cycle is a must that needs to be simulated in any modeled ozone product (all of your six sets are modeled products). Likewise the variability of ozone (including MDA8) is critical in evaluating health/agric. impacts and there needs to be a test of your six 'sets' as to their ability of match extremes.

**Response:**

*This is a very good point. We agree that although the OSDMA8 metric fulfills specific needs for many scientists, regulators, epidemiologists, and policymakers, it certainly is not the only metric of interest. As you have pointed out, metrics capturing the full 24-hour diel cycle of ozone are essential for robust evaluation and validation of ozone from chemical transport models or other models. However, our study is not intended to perform a model evaluation as one would do for a chemical transport model. Rather, we focus here on intercomparison and model evaluation for a single yearly metric (OSDMA8) that is important as a metric adopted by the WHO air quality guidelines, and by the Global Burden of Disease studies. Doing so is necessary because the UKML and NJML datasets estimate monthly average DMA8, and not a finer temporal resolution, and the BME dataset estimates OSDMA8. The chemical renalyses estimate ozone at finer timescales (Table 1), and they have been evaluated comprehensively with respect to observations previously ((Miyazaki et al., 2024; Sekiya et al., 2024; Jones et al., 2024)). We clarify this explicitly in introduction of revised manuscript.*

**Revised:**

*Line 95: "Our study specifically utilizes the OSDMA8 metric because we focus on evaluating long-term ozone exposure, an aspect not comprehensively compared previously among global ozone mapping products."*

*Line 155: "Detailed comparisons of these reanalyses for ozone over the entire troposphere at finer timescales have been conducted by the TOAR-II chemical reanalysis working group (Sekiya et al., 2024; Jones et al., 2024; Miyazaki et al., 2024), but without a focus on the ground level and long-term metric as analyzed here."*

*Line 210: "The OSDMA8 metric is used for long-term ozone exposure given its utility and wide acceptance in health impact studies, despite the inherent loss of shorter temporal dynamics."*

L195ff:  ibid.  This is a mistake to smooth out the fundamental ozone cycles (diel and synoptic).

***Response:***

*Please see the response to the previous comment.*

L220ff:  "We adopted a point-to-grid evaluation approach, where the data from each TOAR-II observation site was matched with a corresponding grid cell in each dataset. For grid cells with a TOAR-II observation but no valid estimate in a dataset (NA value), we used the nearest valid estimate instead."  This seems to ignore the previous TOAR-related work by Schnell where for the high-density of surface sites in EU and N.Am., a 1º×1º grid-cell averaged, hourly surface ozone product was created.

This data set was used to assess extremes and to test the CMIP model's accuracy in seasonal and diel cycle of ozone.  The cell average is the only way to do a fair comparison with the surface sites because of their irregular – sometimes oversampling and sometimes under sampling – many regions.  Comparing surface sites with model cells is dangerous, especially since in this paper their appears to be a lack of understanding of the problems with this approach.  The Schnell data are the obvious choice to validate your six model-data sets, even if it is only for EU and NAm:

doi:10.5194/acp-14-7721-2014

doi:10.5194/acp-15-10581-2015

doi:10.1002/2016GL068060

doi:10.1002/2017GL073044

doi: 10.1073/pnas.1614453114.

Then you can go after the rest of the world (which is very important).

***Response:***

*Thank you for emphasizing this important point. We recognize the significance and validity of the Schnell et al. dataset, which provides 1º×1º grid-cell-averaged hourly ozone data, particularly suitable for analyzing extremes and validating seasonal and diel ozone cycles. However, the goal of our work is to assess the accuracy of gridding products in estimating measured ground concentration point values. The reason for doing this is because the*

*point value is of interest in some applications. For example, exposure scientists are frequently concerned with assessing ozone exposure at specific locations or points. While point values are available at monitoring sites, they are not available away from monitoring sites. Thus, while Schnell's dataset effectively addresses the challenge of spatial representativeness by providing grid-cell ozone values, our focus is explicitly on assessing whether global ozone mapping products can reasonably estimate point-level concentrations at locations lacking monitoring stations. That is why we have adopted this grid-to-point evaluation approach. Any gridded product, such as that of Schnell et al, uses an interpolation of a point value that introduces its own uncertainties and biases. To avoid these additional uncertainties, we directly compared observed point-level ozone values to the nearest available grid estimates. We explicitly acknowledge and clarify this in our methodology part of the revised manuscript.  We have also included a Table (S12) that lists the NA values for each dataset.  The chemical reanalysis datasets at coarse resolution have no NA values. For the other datasets at finer resolution, NA values are mainly along coasts and in the large majority of cases where an NA value exists, an adjacent grid cell is selected for comparison with observations.*

***Revised:***
*Line 239: "Previous research has adopted a 1º×1º grid-cell-averaged hourly ozone data from TOAR observations to evaluate global chemistry model performance over North America and Europe, which is suitable for analyzing extremes and validating seasonal and diel ozone cycles (Schnell and Prather, 2017; Schnell et al., 2015)."*

*Line 243: " We adopted a grid-to-point evaluation approach, where the data from each TOAR-II observation site was matched with a corresponding grid cell in each dataset. For grid cells with a TOAR-II observation but no valid estimate in a dataset (NA value), we used the nearest valid estimate instead."*

*Line 249: " We assessed the performance of each dataset using the coefficient of determination ($R^2$) between ozone estimates and observations, and root mean square error (RMSE) as the primary metrics. We selected the 50 ppb as the threshold for high ozone concentration because it corresponds to the long-term air quality interim target of WHO. These performance metrics should be interpreted considering the spatial representativeness uncertainty which is caused by the grid-to-point evaluation approach."*

This paper is based on comparing 6 different modeled surface ozone dataset with one another and with the TOAR set of surface sites (Table 3).  The comparison of individual sites with grid-cell averages fails to recognize the difficulty of the task and ignores the extensive

efforts to develop unbiased grid-cell means from high-density observations. The authors further corrupt the data set by averaging and smoothing to destroy the fundamental information on ozone variability that is critical for testing the modeled ozone products. The use of these 6 sets, varying in resolution from 0.1 to 2.5 degrees, to map population exposure is premature.

I can not recommend publication of this work as is.

***Response:***

*Thank you for your thoughtful comments. In general, we focus here on an annual metric that is recognized as important for health, which all 6 datasets estimate, whereas the machine learning and BME datasets do not provide estimates at finer temporal resolution. Evaluating fine temporal resolution (daily cycle) is important for typical model evaluations, but that is not the purpose of this study. While the work of Schnell et al is valuable, we do not think it provides a stronger basis for model evaluation than the comparison of individual monitoring sites with grid cell averages from the ozone mapping products, which is a standard method used in our field. Our work is valuable in showing that current products using different methods of estimating ground-level ozone differ from one another and vary in performance against observations, suggesting that further work to better constrain ground level ozone remains important.*

*Reference*

*Jones, D. B. A., Prates, L., Qu, Z., Cheng, W. Y. Y., Miyazaki, K., Inness, A., Kumar, R., Tang, X., Worden, H., Koren, G., and Huijnen, V.: Assessment of regional and interannual variations in tropospheric ozone in chemical reanalyses, 2024.*

*Miyazaki, K., Bowman, K., Marchetti, Y., Montgomery, J., and Lu, S.: Drivers of regional surface ozone bias drivers in chemical reanalysis air quality revealed by explainable machine learning, 2024.*

*Sekiya, T., Emili, E., Miyazaki, K., Inness, A., Qu, Z., Pierce, R. B., Jones, D., Worden, H., Cheng, W. Y., and Huijnen, V.: Assessing the relative impacts of satellite ozone and its precursor observations to improve global tropospheric ozone analysis using multiple chemical reanalysis systems, EGUsphere, 2024, 1-35, 2024.*

---

## Author Comment (AC3)

February 12, 2025

Comments by Owen R. Cooper (TOAR Scientific Coordinator of the Community Special Issue) on:

**Intercomparison of global ground-level ozone datasets for health-relevant metrics**

Hantao Wang, Kazuyuki Miyazaki, Haitong Zhe Sun, Zhen Qu, Xiang Liu, Antje Inness, Martin Schultz, Sabine Schröder, Marc Serre, and J. Jason West

EGUsphere [preprint], https://doi.org/10.5194/egusphere-2024-3723
Discussion started Jan. 3, 2025
Discussion closes Feb. 14, 2025

This review is by Owen Cooper, TOAR Scientific Coordinator of the TOAR-II Community Special Issue. I, or a member of the TOAR-II Steering Committee, will post comments on all papers submitted to the TOAR-II Community Special Issue, which is an inter-journal special issue accommodating submissions to six Copernicus journals:  ACP (lead journal), AMT, GMD, ESSD, ASCMO and BG. The primary purpose of these reviews is to identify any discrepancies across the TOAR-II submissions, and to allow the author teams time to address the discrepancies.  Additional comments may be included with the reviews. While O. Cooper and members of the TOAR Steering Committee may post open comments on papers submitted to the TOAR-II Community Special Issue, they are not involved with the decision to accept or reject a paper for publication, which is entirely handled by the journal's editorial team.

**Comments regarding TOAR-II guidelines:**

TOAR-II has produced two guidance documents to help authors develop their manuscripts so that results can be consistently compared across the wide range of studies that will be written for the TOAR-II Community Special Issue.  Both guidance documents can be found on the TOAR-II webpage: https://igacproject.org/activities/TOAR/TOAR-II

*The TOAR-II Community Special Issue Guidelines*:  In the spirit of collaboration and to allow TOAR-II findings to be directly comparable across publications, the TOAR-II Steering Committee has issued this set of guidelines regarding style, units, plotting scales, regional and tropospheric column comparisons, and tropopause definitions.

*The TOAR-II Recommendations for Statistical Analyses*:  The aim of this guidance note is to provide recommendations on best statistical practices and to ensure consistent communication of statistical analysis and associated uncertainty across TOAR publications. The scope includes approaches for reporting trends, a discussion of strengths and weaknesses of commonly used techniques, and calibrated language for the communication of uncertainty. Table 3 of the TOAR-II statistical guidelines provides calibrated language for describing trends and uncertainty, similar to the approach of IPCC, which allows trends to be discussed without having to use the problematic expression, "statistically significant".

**General comments:**

Line 23
Is there any reason to report 60.8% with one decimal place? Would 61% be better, given the uncertainty in the estimate?
*Response:*
*We agree. 61% is better.*
*Revised:*
*Line 23: "Among the six datasets, the population exposed to over 50 ppb varies from 61% to 99% in East Asia, 17% to 88% in North America, and 9% to 77% in Europe (2006–2016 average)."*
*Line 325: "Regional exposure estimates vary in East Asia, where the proportion of the population exposed to more than 50 ppb ranges from 61% in BME to over 90% in UKML, GEOS-Chem, and TCR-2."*
*Line 420: "In East Asia, exposure levels are consistently higher, with the percentage of the population affected ranging from 61% for BME to more than 90% for UKML, GEOS-Chem, and TCR-2 based on average OSDMA8 data over the same period."*

Line 24
The following statement is not very clear:
"These differences are large enough to impact health and other applications."
I suggest:
"These differences are large enough to impact assessments of health impacts and other applications."
*Response:*
*We agree and have changed to: "These differences are large enough to impact assessments of health impacts and other applications."*
*Revised:*
*Line 25: "These differences are large enough to impact assessments of health impacts and other applications."*

Line 34
Please also provide the uncertainty range, along with the estimate of mortality.

*Response:*
*We add the uncertainty range.*
*Revised:*
*Line 35: "The Global Burden of Disease 2021 (GBD) study estimated that ground-level ozone contributed to approximately 490,000 (95% UI: 107,000–837,000) global deaths in 2021, representing 0.72% (95% UI: 0.16% – 1.18%) of all deaths that year."*

Line 38
Make it clear that these ozone increases refer to ozone above the surface (surface ozone was not reported in this study because the surface observations were from airport runways, which are not representative of typical conditions). When mentioning population-weighted metrics, Gaudel et al. (2020) is not a correct reference as it does not address these metrics. Please provide a different reference. It would be helpful to list some references that provide recent updates on surface ozone trends. One such paper is Chang et al. (2024), submitted to the TOAR-II special issue, which focuses on long-term surface ozone trends across the USA.

*Response:*
*We clarify that tropospheric ozone refers to ozone above the surface and included Chang et al. (2024) as a reference for the surface ozone trend in the U.S.*
*Revised:*
*Line 40: "Gaudel et al. find that since the mid-1990s, tropospheric ozone above the surface has*

*increased across all 11 study regions in the Northern Hemisphere that they defined and analyzed (Western North America, Eastern North America, Southeast North America, Northern South America, Northeast China/Korea, The Persian Gulf, India, Southeast Asia, Malaysia/Indonesia, Europe, Gulf of Guinea) (Gaudel et al., 2020). In the United States, although extreme ground-level ozone concentrations have declined, winter ground-level ozone concentrations have increased in Southwest and Midwest regions since 1990s (Chang et al., 2024)."*

Line 243-244
It is an oversimplification to say that ozone is typically increasing in the northern hemisphere over 2005-2016. First you need to specifically state that you are talking about the OSDMA8 metric, which is very different from the metrics reported by Gaudel et al (2018) and Fleming et al. (2018). These earlier studies showed a range of increasing and decreasing ozone trends that varied by region. The recent trend update by Chang et al. (2024) shows decreasing ozone in the eastern and western USA over the period 2005-2016. I recommend that you refer to studies that have focused on OSDMA8, such as Becker et al., 2023, and Malashock et al., 2022 (see Figure 1 and Figure 2 of Malashock et al., 2022; note that this is the second paper by Malashock, published in 2022; see the reference listed below).

***Response:***
*We delete the sentences reference to Gaudel et al (2018) and Fleming et al. (2018) in this paragraph. Since both Becker et al., 2023, and Malashock et al., 2022 are using the BME dataset, they do not provide independent analysis. We cite Chang et al 2024 in the regional trend comparison part.*

***Revised:***
*Line 277: "Recent analyses using TOAR observations indicate that from 2006 to 2016, most sites in North America experienced decreasing ozone, while many sites in East Asia exhibited significant positive trends (Chang et al., 2024; Fleming et al., 2018; Chang et al., 2017)."*

Line 509
According to the TOAR data use policy (https://toar-data.fz-juelich.de/footer/terms-of-use.html), the TOAR data also needs the following citation:
Schröder et al; TOAR Data Infrastructure;
https://doi.org/10.34730/4d9a287dec0b42f1aa6d244de8f19eb3

***Response:***
*We have added a citation to this reference.*
***Revised:***
*Line 190: "For the evaluation in this project, we utilized both urban and non-urban ground-level ozone observations for the yearly OSDMA8 metric from the updated TOAR-II dataset, covering 2006 to 2016 (Schröder et al., 2021)."*
*Line 527: "Observational data are publicly available from the TOAR-II data portal (last accessed on 15 November 2024, toar-data.org) (Schröder et al., 2021). "*

Figure 1
Following the TOAR-II statistical guidelines, all trends need to be reported with their 95% confidence intervals and *p*-values.
***Response:***
*We added 95% UI to 3 new tables (Tables 2, Table S11, Table S13) presenting trends in the main body and SI, and modified the description.*
***Revised:***
*Line 22: "For example, in Europe, the two chemical reanalyses show an increasing trend while the other datasets show no increase."*
*Line 263: "In Table 2, focusing on the period from 2006 to 2016, we find that NJML is the only dataset*

*showing a downward trend in both area-weighted and population-weighted mean ozone concentrations, with very high certainty. In contrast, TCR-2 and UKML show increasing trends in population-weighted mean ozone during this period with very high certainty."*

*Line 275: "From Table S11, we observe that some regions exhibit a clearer trend from 2006 to 2016, with very high certainty across six datasets. In East Asia, BME and NJML observe decreasing trends, whereas the other 4 datasets display increasing trends. In North America, all datasets display a downward trend, and in Europe, BME, NJML, UKML and TCR-2 show a decline, contrasting with increases in CAMS and GEOS-chem  datasets. Recent analyses using TOAR observations indicate that from 2005 to 2016, most of North America sites experienced decreasing ozone, while many sites in East Asia exhibited significant positive trends."*

*Line 415:  "NJML demonstrates a decreasing trend in global population-weighted and area-weighted yearly mean over the 2006-2016 period, while the five others exhibit either increasing trends or no clear trend."*

*Line 499: "Regionally, all datasets show a downward trend in North America, and only BME and NJML datasets demonstrate a downward trend in East Asia; In Europe, BME, UKML, NJML and TCR-2 report a downward trend, while the other two chemical reanalysis datasets reveal an upward trend that is not seen in observations."*

Figure 7
These figures need to be reoriented, with the TOAR-II observation being the independent variable on the x-axis, and the model output being the dependent variable on the y-axis.
***Response:***
*We change Figure 7 to show the TOAR-II observations on the x-axis.*

**References**

Chang, K.-L., McDonald, B. C., and Cooper, O. R. (2024), Surface ozone trend variability across the United States and the impact of heatwaves (1990–2023), EGUsphere [preprint], https://doi.org/10.5194/egusphere-2024-3674 (submitted to ACP as a contribution to the TOAR-II Community Special Issue)

Malashock, Daniel A., Marissa N. Delang, Jacob S. Becker, Marc L. Serre, J. Jason West, Kai-Lan Chang, Owen R. Cooper, Susan C. Anenberg (2022), Global Trends in Ozone Concentration and Attributable Mortality for Urban, Peri-Urban and Rural Areas between 2000 and 2019: A Modelling Study, The Lancet Planetary Health, Volume 6, Issue 12, Pages E958-E967, https://doi.org/10.1016/S2542-5196(22)00260-1

---

## Author Response (AR2)

Reviewer Comments: Black font

Author Responses: *Black italics*

Manuscript Revisions: Blue italics

**Response to Reviewer #1**

1. Re-reviewing this manuscript with the authors' responses to the first review does not provide any more encouragement that the paper adds significant new knowledge to the atmospheric chemistry domain. In terms of regulatory issues, it may be fine, but the extreme limitation of their diagnostics makes the paper useless for diagnosing the problems with modeling surface ozone. Further, calculating human impacts from 6-month OSDMA8 ozone when you show the incredible biases of the modeled ozone seems like going too far, e.g., the numbers in Table 3 have no uncertainties related to the obvious model bias, so how can they be published? "The ozone seasonal daily maximum 8-hour average mixing ratio " = six-month running monthly mean). This diagnostic totally obscures all issues of extremes and even hides the seasonal cycle, removing most of the inter-month variability. The use of a six-month DMA8 because some of the models only did that is still a poor excuse. Not looking at the diel cycle and extreme days means you cannot understand why the models fail to match DMA8. The CAMS and TCR-2 results seem sensible (2-hr and 3-hr ozone) and at a more reasonably resolution to make a useful comparison.

**Response:**

We thank Reviewer #1 for reviewing our manuscript a second time, and for their thoughtful comments. We have made significant changes to the paper in response to reviews. These revisions include:

- 1) We have changed the order of Sections 4 and 5 in the manuscript, and have modified several results to include uncertainty. By reordering these sections, we discuss the performance evaluation of the different datasets with respect to measurements first, and then include estimates of uncertainty from the performance evaluation in results for the inter-comparison of the different datasets.
- 2) As Reviewer #1 suggested, we now add a grid-to-grid comparison method in addition to the grid-to-point method used in earlier versions of the manuscript. The grid-to-grid method follows from the methods of Schnell et al. (2015)

In this comment, Reviewer #1 recommends that the paper be rejected since our use of an aggregate annual ozone metric obscures diurnal and seasonal cycles, and therefore the evaluation with respect to observations is inadequate. We agree that for some air quality related research, evaluating these short-term timescales is essential. However, the primary goal of this paper is to intercompare global datasets specifically for their use in long-term health exposure

related studies. For this purpose, the 6-month seasonal daily maximum 8-hour average (OSDMA8) ozone is a standard metric, as used by the Global Burden of Disease (GBD) and in the World Health Organization (WHO) Air Quality Guidelines. This metric is designed to capture the long-term exposure during these peak seasons which is most relevant for assessing long-term health impacts. In contrast, short-term exposure studies use daily or hourly metric to investigate the health risk related to acute air pollution events. Reviewer #1 also suggests that we conduct a traditional model evaluation for only the chemical reanalyses, since they report 2- or 3-hourly ozone. But such comparisons with observations have already been published (Jones et al., 2025; Miyazaki et al., 2025) and this is not the point of this paper. Rather, we compare these datasets for an annual metric OSDMA8 that is used by GBD and in the WHO Air Quality Guidelines. In doing so, we aim to compare datasets of global ground-level ozone for this health-focused metric, that have been generated by different methods – geostatistical data fusion, machine learning, and chemical reanalyses. Estimating how biases in these available ozone mapping products lead to systematic differences in exposure estimates is a significant and important topic for the community. Because these datasets have not been compared systematically, our work makes a direct contribution to this ongoing challenge. We have carefully considered Reviewer #1's suggestion regarding using a short-term metric like 2-hr and 3-hr ozone for intercomparison but addressing this would require a fundamental change to the purpose and scope of this paper, which we believe would not be appropriate here.

Regarding Reviewer #1's comments about uncertainty, we have made substantial changes to the paper. We have changed the order of Sections 4 and 5 to present dataset evaluation with respect to observations first, before the dataset intercomparison. We have also used estimates of uncertainty from the evaluation section (now Section 4) to quantify uncertainties that are now presented in the intercomparison section (now Section 5). We thank Reviewer #1 for these valuable comments. In this way, we expand our discussion to provide clearer interpretations of how the uncertainties from evaluation impact on the paper's main findings. With these interpretations, we draw more meaningful conclusions.

2. Looking at these two responses shows that there is little understanding of the problem: "However, the goal of our work is to assess the accuracy of gridding products in estimating measured ground concentration point values.

"That is why we have adopted this grid-to-point evaluation approach. Any gridded product, such as that of Schnell et al, uses an interpolation of a point value that introduces its own uncertainties and biases. To avoid these additional uncertainties, we directly compared observed point-level ozone values to the nearest available grid estimates.

You did not "avoid" the uncertainties with the problem that Schnell dealt with, you simply ignored them. There is nothing here that attempts to deal with the problems of creating a high-resolution map based on the site measurements and then integrating it to get the cell average.

This problem is what Schnell's first paper spent most of its effort on. I may have missed it but I find no serious effort to optimize the point-area comparison.

"Previous research has adopted a 1°×1° grid-cell-averaged hourly ozone data from TOAR observations to evaluate global chemistry model performance over North America and Europe, which is suitable for analyzing extremes and validating seasonal and diel ozone cycles (Schnell and Prather, 2017; Schnell et al., 2015)."

"We adopted a grid-to-point evaluation approach, where the data from each TOAR-II observation site was matched with a corresponding grid cell in each dataset. For grid cells with a TOAR-II observation but no valid estimate in a dataset (NA value), we used the nearest valid estimate instead."

Minor point – Schnell gathered all the AQ station data in N.Am. and EU, not specifically the TOAR data. Really, a "grid-to-point evaluation" is simply saying that every point in a cell should have the same value as the mean of that cell. This is quite apparently false when you have several nearby sites. The algorithm here (last sentence) does not really address how "far" a single station can reach? or why? "nearest" valid estimate is not provide a scientific comfort level. "our focus is explicitly on assessing whether global ozone mapping products can reasonably estimate point-level concentrations at locations lacking monitoring stations.

I cannot see how you can begin to do that without models that resolve station-to-station differences (1 km) and so you really cannot say this.

**Response:**

Reviewer #1 questions our use of a grid-to-point comparison of observations with gridded ozone estimates, suggesting that we spatially average observations to a grid before comparing with the six ozone datasets. Such grid-to-point comparison methods are widely used in our field, and the specific dataset developed by Schnell et al. is limited to North America and Europe and so does not match the spatial or temporal scope of our study. We thank Reviewer #1 pointing out this concern, and we agree with Reviewer #1 that these issues pose a challenge when evaluating coarse-resolution datasets against observations. To address this issue, we have implemented two evaluation methods in our revised manuscript.

- 1. Grid-to-Grid evaluation: We have re-gridded the TOAR-II observations onto the native grid resolution of each dataset. In doing so we use methods similar to those of Schnell et al. (2015), but here we gridded observations for the whole TOAR-II dataset globally and for more years than had been done previously. This method addresses the representativeness issue by comparing the value of re-gridded observational grid cell to the value of dataset's grid cell for the same spatial resolution.
- 2. Grid-to-Point evaluation: This is the traditional method used in the original manuscript where the value of the dataset's grid cell is compared to all observations within that cell.

This method can ensure that all evaluations are the same sample size given by the number of observations.

We thank Reviewer #1 for their comments that led us to include the grid-to-grid methods, which we feel strengthened the manuscript.

We have revised the methods as follows:

Line 214: Considering that the six datasets have different resolutions and are designed for different applications, we adopted a dual evaluation strategy to provide a comprehensive assessment of their performance. The first method is a grid-to-grid evaluation. Similar to the approach of Schnell et al. (2015), we re-gridded TOAR-II observations to a 0.1° x 0.1° resolution by an inverse distance weighted method and then aggregated them to match the native resolution of each of the six datasets. In this approach, the sample size for each evaluation varies reflecting the varying resolution of the datasets; for 2016, BME had 173,718 grid cell pairs, NJML had 7,099, UKML had 162,419, CAMS had 4,614, GEOS-Chem had 782, and TCR-2 had 2,195. We also adopted the grid-to-grid evaluation method for regional evaluations, as it provides better spatial representativeness over large areas. To quantify the uncertainty of the six datasets' estimates, we determined the lower and upper bounds (95% confidence interval), derived from the grid-to-grid regression analysis performed between the TOAR-II observations and each of the six datasets at their native resolutions.

Line 224: The second method is a standard grid-to-point evaluation. This approach ensures a consistent sample size across all datasets by comparing each dataset's estimate at the grid cell containing an observation location. For grid cells containing a TOAR-II site but no valid estimate (NA value), we used the nearest valid estimate instead. This method captures a penalty for missing data and coarse resolution, only BME, NJML, and UKML had a small number of missing estimates at TOAR-II locations. The grid-to-point method was used to evaluate model bias, as it ensures a consistent sample size across all datasets when performing evaluations on different quantiles of the TOAR-II observations.

By presenting results from both methods in section 4.1, we provide a more robust and comprehensive assessment of dataset performance. Overall, our main conclusions and the relative performance rankings of the datasets remain largely consistent across both evaluation methods. As expected, the grid-to-grid approach generally results in lower RMSE and higher R² values, as it averages out localized errors that are more prominent in the direct grid-to-point evaluation. However, the difference of two methods does not change the fundamental takeaways of our analysis.

We have updated the results throughout the manuscript to describe the evaluation from both the grid-to-grid and grid-to-point methods. The following are some key passages from the manuscript, updated to reflect the revisions:

Line 275: For 2016, BME outperforms other datasets in both evaluation method, with the highest  $R^2$  (0.75 for grid-to-grid, 0.63 for grid-to-point) and lowest RMSE (4.25 ppb for grid-to-grid, 5.28 ppb for grid-to-point), its density cores intersecting the y=x line.

Line 280: In Fig. 1(a), all three chemical reanalysis datasets exhibit a moderate  $R^2$  ranging from 0.51 to 0.60 for grid-to-grid and 0.35 to 0.41 for grid-to-point, comparable to the performance of the machine learning datasets, which have  $R^2$  values of 0.50 and 0.56 for grid-to-grid, 0.37 and 0.38 for grid-to-point. Among these five datasets, CAMS has the lowest RMSE (6.00 ppb for grid-to-grid and 7.59 ppb for grid to point), which is better than other chemistry reanalysis products but relatively low  $R^2$  (0.51 for grid-to-grid and 0.35 for grid-to-point). Its density cores slightly below the y=x line suggests CAMS estimates are marginally lower than TOAR-II observations. GEOS-Chem and TCR-2 demonstrate adequate performance, albeit with higher RMSE values of 8.47 ppb and 10.26 ppb for grid-to-grid, 10.27 ppb and 13.23 ppb for grid-to-point, respectively.

Line 362: In grid-to-grid evaluation, GEOS-Chem shows an overall better performance in  $R^2$  than CAMS, TCR-2 and UKML.

Line 369: From 2006 to 2016, the performance rankings derived from  $R^2$  values varied significantly between the two evaluation scenarios, whereas the RMSE based rankings were nearly consistent.

Line 550: In instances of missing model estimates, we default to the nearest valid estimate to evaluate with TOAR-II observations or re-gridded grid cell. For datasets with coarse spatial resolution, this method may increase or reduce bias by double counting.

3. The authors have done nothing to address the primary problems with this analysis: the use of 6-month averages of MDA8 to compare with models; and the fundamental methodology of how to compare a mean grid-cell value with point measurements, especially when there are several in a cell. The latter is clearly major unresolved problem (here at least). What happens if you have three different sites within a cell, each with three different values – how does one compare and derive R2? I realize that the authors may not be able to do this given the material they have, and thus this paper belongs in an air quality management journal that deals with meeting regulations, not in a science journal like ACP.

**Response:**

Please see our responses above to (#1) the overall purpose and scope of the paper, and (#2) the evaluation method.

Reviewer #1 argues that our paper is outside of the scope of ACP. We acknowledge Reviewer #1's concern regarding the scope of ACP. However, we strongly believe that our paper fits well within the journal's scope for the following reasons:

- 1. The datasets we evaluate and compare are of significant interest to the ACP readership, which includes atmospheric scientists, climate modelers, and policy makers interested in the impacts of ozone exposure on health, agriculture, and ecosystems. Therefore, our work is directly relevant to the ACP community.
- 2. Our findings of significant differences among the datasets despite the fact that some methods use some of the same inputs, which has not been shown previously, highlights the importance of continued research on global ozone distributions. Machine learning and other methods have not converged on a single correct global ozone distribution. This is a direct contribution to atmospheric research community.
- 3. This paper was submitted as part of the TOAR-II special issue, and our work has been presented at TOAR-II online meetings, as well as at the CMAS conference. Our work has been received enthusiastically, with colleagues commenting both that it was conducted thoroughly, and that such a systematic comparison of different datasets that are used widely is overdue and important.

**Response to Reviewer #2**

The authors have addressed some of the reviewers' suggestions, and there are some improvements in the revised manuscript. However, several important issues remain and require further revision. Failure to address the comments from the first round seriously compromises the value of this study. Unless these critical points are properly dealt with, the contribution of the work remains questionable.

**Response:**

We thank Reviewer #2 for their careful reading of the manuscript and for providing critical feedback. We agree with the reviewer that our revised manuscript did not sufficiently address the comments in first round. We have thoroughly revised the manuscript to address these critical points.

1. Impact of data uncertainty on related analysis and reorganizing structure (i.e. third comment in the 1st round): Regarding this issue, I do not agree with your response. While readers can draw their own conclusions, it is the authors' fundamental responsibility to provide clear and specific interpretations based on the findings. Leaving key aspects of the analysis entirely up to the reader undermines the completeness and clarity of the work. This approach also deviates from the stated aims and title of the study, which emphasize evaluating uncertainty and accuracy through comparisons between observation and datasets. One of the core values of this research should be to account for these uncertainties and to draw meaningful conclusion – particularly about implications for public health and agriculture. Moreover, Sections 4 and 5 lack a logical flow, which further weakens the clarity and impact of the manuscript. Without a through and well-reasoned interpretation of the results (i.e., the uncertainty and its impact on trend analysis and

ozone exposed population assessment), the manuscript does not fulfill its stated objectives and, in its current form, is not suitable for acceptance in ACP.

**Response:**

Thank you for this valuable suggestion to improve the manuscript's flow, and we apologize for this oversight in the first revision. Now we have reversed the order of sections 4 and 5, to present dataset evaluation with respect to observations first, before the dataset intercomparison. To directly address your concern, we have also used the results of the model evaluation to quantify uncertainty in some of the results presented in the dataset intercomparison (now Section 5). Specifically, for the exposure assessment (now in Section 5.4), we add the lower and upper bound of population exposure to the OSDAM8 level in Supplementary Figure S12. In Table 3, for share of population exposure to different ozone thresholds, we now add the lower and upper bound based on the evaluation results.

*In addition to changing the order of Sections 4 and 5, we have also changed the text as follows:*

Line 266: To quantify the uncertainty in our exposure analysis, we established lower and upper bounds for all population exposure and share of population estimates. The OSDMA8 95% confidence interval (CI) for each dataset is determined through a grid-to-grid linear regression between each dataset and the re-gridded TOAR-II observations based on  $0.1^{\circ} \times 0.1^{\circ}$ grid cells.

Line 438: We also calculated the distribution of population regarding the lower and upper bounds of OSDMA8 from 2006 to 2016 for each dataset, as shown in Figure S12.

Line 452: Results are presented as the estimate with the lower and upper bound in parentheses (e.g., 42% [24%, 66%]). Detailed table of population share for each year (2006 to 2016) are shown in Table S10.

We have also substantially expanded our discussion to provide a clearer interpretation of how the evaluation results impact on the paper's main findings. For example, the ozone trend analysis has been revised to include how systematic biases found in our evaluation (some datasets are overestimate) contribute to the divergent long-term trends among datasets. With these interpretations, we draw more meaningful conclusions.

Line 471: In addition, for chemical reanalysis datasets, there is a clear trade-off between capturing the spatial pattern and the accuracy. As shown in Fig. 2, TCR-2, GEOS-Chem all have widespread overestimation, but they often capture spatial patterns more effectively (higher R²). Conversely, CAMS exhibits low bias in RMSE but shows worse spatial correlation in China. All six datasets show a reduced performance at higher ozone concentrations (>50 ppb), which may complicate their accuracy for assessing long term high-pollution exposure. Furthermore, most datasets perform better in regions with lower monitoring density (e.g., the United States and China) than in those with higher density (e.g., Japan and South Korea), which suggests that resolving high-resolution local ozone distributions remains challenging even with a good amount

of observational data. The performance of each dataset impacts the accuracy of trend analysis (Fig. 5 and Fig. 6) and population exposure assessment (Fig. 10), shown as uncertainty in these Figures, which may lead to different results when compared to the WHO guideline and interim target.

Line 490: From the comparison, the large disagreements among the six datasets regarding ozone trends, population exposure, and concentration estimates are a direct consequence of the systematic biases and performance issues identified in the evaluation.

Line 497: These uncertainties critically undermine the reliability of population exposure assessment.

Line 503: More importantly, the evaluation reveals that all datasets perform poorly at high ozone levels (> 50 ppb). This highlights the importance of removing systematic biases from these data sets before applying them to exposure estimates.

Line 509: And from the evaluation, we find that all datasets perform well in the United States, which makes the downward trend more reliable.

Line 570: Regionally, all datasets show a downward trend in North America, and the evaluation results make this trend more reliable. Only BME and NJML datasets demonstrate a downward trend in East Asia, and they also fit well with TOAR-II observations in population density distribution.

Line 579: The coarse-resolution datasets, GEOS-Chem and TCR-2, perform well in grid-to-grid evaluations at their native resolutions, making them effective for studying long-term regional ozone effects. However, because of their coarser resolutions, these two datasets cannot capture site-level distributions and exhibit greater bias than the higher-resolution BME, CAMS, and NJML datasets. UKML, despite its relatively fine resolution (0.125°), shows larger biases and a lower R2. The superior performance of BME and NJML should be noted with the fact that both datasets use observational data for input or training, which gives them an inherent advantage in these evaluations.

2. Again, regarding the fourth comment in the first round, I don't think the authors thoroughly evaluate the performance of the datasets. The authors missed their scientific discussion and the implications.

**Response:**

We thank the reviewer for this excellent suggestion. We agree that a direct comparison of population exposure at the observation sites provides a more thorough evaluation and strengthens the paper's scientific discussion. We have now done this, but we are limited in doing this to the 3 world regions with a density of observation sites. We re-gridded TOAR-II observations to a  $0.1^{\circ}$  x  $0.1^{\circ}$  resolution by an inverse distance weighted method. The results are

now presented in Figure 3 and discussed in Section 4.2. We use these results to draw clearer conclusions about the implications for regional exposure studies and to better identify the uncertainty in the following exposure assessments. We also present these results for the lower and upper bounds for the population exposure estimates, given uncertainty based on the evaluation of each dataset in Figure S12. We think that this is a valuable addition to the paper, so thank you for the suggestion.

The text is revised as follows:

Figure 3: Population-weighted exposure distributions for OSDMA8 in 2016 in three regions: East Asia (EAS), Europe (EUR), and North America (NAM) (regions defined in Table S7). Each panel compares the distribution derived from the TOAR-II observations (black line) with estimates from six datasets (colored lines), calculating the population-weighted kernel density estimate, only for grid cells where TOAR-II measurements exist.

Line 317: Fig. 3 presents the distribution of population exposure calculated from six datasets and the gridded TOAR-II observations in three world regions with a high density of observations, for 2016. Here we calculate the population-weighted kernel density for population exposure to OSDMA8 concentrations, based on the  $0.1^{\circ} \times 0.1^{\circ}$  resolution for each region, only for grid cells where the re-gridded TOAR-II data have a value. Corresponding plots for other years (2006 to

2015) are shown in Figure S6. Overall, the datasets are widely distributed, and the estimated exposure peaks vary. In East Asia (EAS), the population is exposed to high ozone concentrations. The concentration distribution is broad and has multiple peaks from TOAR-II observations, indicating a complex pollution environment, with a large population exposed to concentrations frequently exceeding 50 ppb, even 70 ppb. BME and NJML show a similar distribution as TOAR-II. Significant differences exist between UKML, CAMS and GEOS-Chem with the TOAR-II data for EAS. In Europe (EUR), exposure is concentrated between 40 and 50 ppb, indicating a more moderate and uniform exposure. The BME and CAMS have the best fit with the TOAR-II. NJML, UKML, GEOS-Chem, and TCR-2 show a peak at a higher ozone concentration range of 50–60 ppb. In North America (NAM), exposure peaks sharply in the 40 to 50 ppb range, which is slightly higher and more concentrated than in Europe. The NJML dataset agrees best with the shape of the TOAR-II distribution, and GEOS-Chem and BME capture the overall shape of the major exposure peaks well.

Line 440: Populations in regions such as East Asia and South Asia appear to be exposed to higher ozone concentrations in all datasets compared to other regions, which supports our findings from exposure based on TOAR-II observations in Fig. 3.

3. The numbering of the supplementary table (Table S) is not presented in sequential order, which can cause confusion. Ensuring correct and consistent numbering of all tables and figures is a basic requirement that should be addressed prior to submission. Please revise the manuscript carefully to ensure that all Tables (S) and Figures (S) are accurate and properly ordered.

**Response:**

We thoroughly reviewed the supplementary information, and ensure all tables and figures are now numbered accurately and appear in sequential order in main paper.

4. Units are missing in Table 2. For clarity and scientific accuracy, please add the relevant units to all applicable columns or data entries.

**Response:**

We thank the reviewer for catching this oversight. We have now added the appropriate units (ppb/year) for the trend slope and its confidence interval. Please note that this table is now Table 3 in the revised manuscript.

5. Even in the revised version, some statements still lack objective explanations based on consistent criteria. For example, the manuscript highlights a downward trend only for NJML (in Line 264); however, BME also shows a concurrent downward trend in both area- and population-weighted metrics. If the authors interpret the populated-weighted trend of BME (i.e., -0.04) as

insufficient to indicate a clear trend, then by the same standard, none of the remaining five datasets show an increasing trend either. In this context, the statement in line 415 needs to be revised. More importantly, the manuscript lacks a clear definition of what constitutes a "clear" or "unclear" trend. The criteria used to make such classifications should be explicitly stated—whether based on slope magnitude, statistical significance, or another method. Without a consistent and objective basis for trend interpretation, the analysis risks appearing arbitrary and subjective. Please review the manuscript carefully to ensure that all claims are supported by clear, objective, and consistently applied analytical reasoning. In addition, the authors are advised to carefully review all statements throughout the manuscript.

**Response:**

We agree that our initial manuscript lacked a clear and consistent definition for interpreting trends, which led to the issues you identified. To fix this issue, we explicitly define our criteria for trend and certainty in the Methods section (section 3). The interpretation is based on the statistical significance (p-value) of the trend's slope, categorized by levels of certainty (e.g., very high, low, etc.).

Line 255: We calculated the yearly ozone trend using 50% quantile regression for each dataset using both population-weighted and area-weighted approaches, with details of the calculation methods provided in Text S2. In this study, the trend is interpreted from the slope of the quantile regression, and confidence in the trend is determined by its p-value:  $p \le 0.01$  is considered very high certainty; 0.01 , high certainty; <math>0.05 , medium certainty; <math>0.1 , low certainty; and <math>p > 0.33, no evidence.

Line 266: To quantify the uncertainty in our exposure analysis, we established lower and upper bounds for all population exposure and share of population estimates. The OSDMA8 95% confidence interval (CI) for each dataset is determined through a grid-to-grid linear regression between each dataset and the re-gridded TOAR-II observations based on  $0.1^{\circ} \times 0.1^{\circ}$ grid cells.

Using these objective criteria, we have carefully revised all trend-related statements throughout the manuscript for consistency. For the specific example you noted, the text now clarifies that while both **NJML** and **BME** show downward trends, they do so with different levels of statistical confidence ('very high certainty' vs. 'low certainty').

*The text is revised as follows:*

Line 382: In Table 3, focusing on 2006 to 2016, we find that NJML was the only dataset to exhibit a downward trend with very high certainty for both area- and population-weighted mean ozone concentrations. In contrast, TCR-2 and UKML only show increasing trends in population-weighted mean ozone during this period with very high certainty. However, while the BME

dataset shows a negative slope for the area-weighted mean, this downward trend has only low certainty; for the population-weighted mean, there is no evidence of a decreasing trend.

Line 477: Despite this, the three chemical reanalysis datasets unexpectedly outperform the machine learning datasets in R2 (TCR-2, GEOS-Chem) and in RMSE (CAMS) over the full year 2016.

Line 493: NJML demonstrates a very high certainty decreasing trend in global population-weighted and area-weighted yearly mean over the 2006-2016 period. While TCR-2 and UKML exhibit very high certainty increasing trends in global population-weighted mean which relates to their overestimation.

6. (Sect 3.3, Line 224-225) Regarding 6th comment in the 1st round, the concern is not about missing data. Of course, reanalysis datasets like GEOS-CHEM, CAMS, and TCR-2 have complete spatial coverage. The issue is the lack of spatial representativeness when comparing coarse-resolution grid cells with single (or multiple) observation sites. A single or (multiple) monitoring station(s) may not adequately represent the entire of a coarse grid cell. The authors need to justify how the address this potential mismatch in spatial representativeness.

**Response:**

We thank the reviewer #2 for pointing out this concern regarding the potential mismatch in spatial representativeness. We agree this is a challenge when evaluating coarse-resolution datasets against observations. To address this issue, we have implemented two evaluation scenarios in our revised manuscript.

- 1. Grid-to-Grid evaluation: We have re-gridded the TOAR-II observations onto the native grid resolution of each dataset. In doing so we use methods similar to those of Schnell et al. (2015), but here we gridded observations for the whole TOAR-II dataset globally and for more years than had been done previously. This method addresses the representativeness issue by comparing the value of re-gridded observational grid cell to the value of dataset's grid cell for the same spatial resolution.
- 2. Grid-to-Point evaluation: This is the traditional method used in the original manuscript where the value of the dataset's grid cell is compared to all observations within that cell. This method can ensure all evaluations are the same sample size given by the number of observations.

By presenting results from both methods in section 4.1, we provide a more robust and comprehensive assessment of dataset performance. We have revised as follows:

Line 214: Considering that the six datasets have different resolutions and are designed for different applications, we adopted a dual evaluation strategy to provide a comprehensive assessment of their performance. The first method is a grid-to-grid evaluation. Similar to the approach of Schnell et al. (2015), we re-gridded TOAR-II observations to a 0.1° x 0.1° resolution by an inverse distance weighted method and then aggregated them to match the native resolution of each of the six datasets. In this approach, the sample size for each evaluation varies reflecting the varying resolution of the datasets; for 2016, BME had 173,718 grid cell pairs, NJML had 7,099, UKML had 162,419, CAMS had 4,614, GEOS-Chem had 782, and TCR-2 had 2,195. We also adopted the grid-to-grid evaluation method for regional evaluations, as it provides better spatial representativeness over large areas. To quantify the uncertainty of the six datasets' estimates, we determined the lower and upper bounds (95% confidence interval), derived from the grid-to-grid regression analysis performed between the TOAR-II observations and each of the six datasets at their native resolutions.

By presenting results from both methods in section 4.1, we provide a more robust and comprehensive assessment of dataset performance. Overall, our main conclusions and the relative performance rankings of the datasets remain largely consistent across both evaluation methods. As expected, the grid-to-grid approach generally results in lower RMSE and higher  $R^2$  values, as it averages out localized errors that are more prominent in the direct grid-to-point evaluation. However, the difference of two methods does not change the fundamental takeaways of our analysis.

We have updated the results throughout the manuscript to describe the evaluation from both the grid-to-grid and grid-to-point methods. The following are some key text from the manuscript, updated to reflect the revisions:

Line 275: For 2016, BME outperforms other datasets in both evaluation method, with the highest  $R^2$  (0.75 for grid-to-grid, 0.63 for grid-to-point) and lowest RMSE (4.25 ppb for grid-to-grid, 5.28 ppb for grid-to-point), its density cores intersecting the y=x line

Line 280: In Fig. 1(a), all three chemical reanalysis datasets exhibit a moderate  $R^2$  ranging from 0.51 to 0.60 for grid-to-grid and 0.35 to 0.41 for grid-to-point, comparable to the performance of the machine learning datasets, which have  $R^2$  values of 0.50 and 0.56 for grid-to-grid, 0.37 and 0.38 for grid-to-point. Among these five datasets, CAMS has the lowest RMSE (6.00 ppb for grid-to-grid and 7.59 ppb for grid to point), which is better than other chemistry reanalysis products but relatively low  $R^2$  (0.51 for grid-to-grid and 0.35 for grid-to-point). Its density cores slightly below the y=x line suggests CAMS estimates are marginally lower than TOAR-II observations. GEOS-Chem and TCR-2 demonstrate adequate performance, albeit with higher RMSE values of 8.47 ppb and 10.26 ppb for grid-to-grid, 10.27 ppb and 13.23 ppb for grid-to-point, respectively.

Line 362: In grid-to-grid evaluation, GEOS-Chem shows an overall better performance in  $R^2$  than CAMS, TCR-2 and UKML.

Line 369: From 2006 to 2016, the performance rankings derived from  $R^2$  values varied significantly between the two evaluation scenarios, whereas the RMSE based rankings were nearly consistent.

Line 550: In instances of missing model estimates, we default to the nearest valid estimate to evaluate with TOAR-II observations or re-gridded grid cell. For datasets with coarse spatial resolution, this method may increase or reduce bias by double counting.

7. The amount of supplementary material is excessive and may overwhelm the reader. The authors are encouraged to condense and streamline the supplementary information by summarizing key findings and removing redundancies. Presenting only the most relevant and necessary content will improve the clarity and accessibility of the supporting materials. In particular, Figure S4 does not appear to add meaningful value.

**Response:**

We have revised the supplement to remove some tables and figures that do not add much meaningful value. The specific tables and figures removed are as follows, based on their original numbering:

- Table S12. Number of NA value at all TOAR-II stations of six datasets in 2016.
- Table S13. Yearly trends of area weighted, and population weighted global mean of ground-level ozone for six datasets with 95% confidence intervals and p-values over full time period of each dataset.
- Figure S4. Population weighted ozone (OSMDA8) trends per decade for six datasets, calculated over the full period of each dataset.
- Figure S8. Heatmaps of pairwise correlation (Pearson R) between each dataset from 2006 to 2016.
- Figure S9. Heatmaps of pairwise Root mean square difference (RMSD) between each dataset from 2006 to 2016.
- Figure S10. Population-weighted ozone (OSMDA8) for each year from 2006 to 2016 in different regions. The horizontal axis represents ozone exposure concentrations, and the vertical axis represents population size.
- Figure S12. Performance evaluations of six datasets with TOAR-II observations for OSDMA8 for each year from 2006 to 2015. The evaluation only includes monitor stations above 50ppb in TOAR-II network for each year (2006 to 2015).